# An increase in the spatial extent of European floods over the last 70 years

Beijing Fang[1], Emanuele Bevacqua[1], Oldrich Rakovec[2,3], and Jakob Zscheischler[1,4]

[1]Department of Compound Environmental Risks, Helmholtz Centre for Environmental Research - UFZ, Permoserstr. 15, 04318 Leipzig, Germany

[2]Department of Department Computational Hydrosystems, Helmholtz Centre for Environmental Research - UFZ, Permoserstr. 15, 04318 Leipzig, Germany

[3]Faculty of Environmental Sciences, Czech University of Life Sciences Prague, Praha-Suchdol 16500, Czech Republic

[4]Technische Universität Dresden, Dresden, Germany

**Correspondence:** Beijing Fang (beijing.fang@ufz.de)

**Abstract.** Floods regularly cause substantial damage worldwide. Changing flood characteristics, for instance due to climate change, pose challenges to flood risk management. The spatial extent of floods is an important indicator for potential impacts, as consequences of widespread floods are particularly difficult to mitigate. The highly uneven station distribution in space and time, however, limits the ability to quantify flood characteristics, and in particular changes in flood extents, over large regions.

5 Here, we use observation-driven routed runoff simulations over the last 70 years in Europe from a state-of-the-art hydrological model (mHM) to identify large spatio-temporally connected flood events. Our identified spatio-temporal flood events compare well against an independent flood impact database. We find that flood extents increase by 11.3 % on average across Europe. This increase occurs over most of Europe, except for parts of eastern Europe and southwestern Europe. Over northern Europe, the increase in flood extent is mainly driven by the overall increase in flood magnitude caused by increasing precipitation and

10 snowmelt. In contrast, the increasing trend in flood extent over central Europe can be attributed to an increase in the spatial extent of heavy precipitation. Overall, our study illustrates the opportunities of combining long-term consistent regional runoff simulations with a spatio-temporal flood detection algorithm to identify large-scale trends in key flood characteristics and their drivers. The detected change in flood extent should be considered in risk assessments as it may challenge flood control and water resource management.

15 *Copyright statement.* TEXT

# 1 Introduction

River floods are a hazard that can endanger lives, damage properties and seriously disrupt communities. Overall, river floods cause annual global losses of around $104 billion US and have the largest population exposure compared to other natural hazards (United Nations Office for Disaster Risk Reduction, 2015). Due to large-scale weather conditions such as widespread precipitation extremes (Bevacqua et al., 2021) and certain landscape properties, when a river experiences a flood, the surrounding rivers might experience floods simultaneously (Berghuijs et al., 2019a). Such widespread river floods might cause overall impacts that could surpass the sum of individual localized flood events (Zscheischler et al., 2018, 2020); in particular when limits for emergency response, disaster rescue, and insurance payouts are exceeded (Kemter et al., 2020; Jongman et al., 2014). For instance, in July 2021, devastating floods induced by extreme rainfall swept through several countries across Europe including Austria, Belgium, Croatia, Germany, Italy, Luxembourg, the Netherlands, and Switzerland. In total, the floods took over 200 lives and caused damages worth more than $54 US billion (Tradowsky et al., 2023). Events such as this illustrate the importance of assessing spatially connected river floods to improve risk analysis and flood risk management today and in future.

So far, previous research has predominantly examined flood events based on a single basin or region perspective. Studies focusing on flood extent are often confined to a limited number of events or regions (Blöschl et al., 2013; De Luca et al., 2017; Kundzewicz et al., 2013; Merz et al., 2018; Stadtherr et al., 2016) and approaches that allow for quantifying the spatial extent of floods across multiple basins or regions are limited (Berghuijs et al., 2019a). To fill this gap, Berghuijs et al. (2019a) proposed the concept of "flood synchrony scale" to measure the spatial scale of cross-basin floods. Furthermore, Brunner et al. (2020) and Brunner and Fischer (2022) used "flood connectedness" to identify the spatial dependence of flood events over remote regions. Based on these concepts, factors and driving processes that influence the spatial synchrony scale and spatial dependence of floods could be explored. Localized conditions (e.g., topography, soil type), as well as driving mechanisms (e.g., rainfall-driven or snowmelt-induced flood), were found to influence the spatial extent and dependence of floods (Berghuijs et al., 2019a; Brunner and Fischer, 2022; Kemter et al., 2020).

Despite the advances revealed by these pioneering studies, their conclusions rely on observed data from station measurements. Consequently, their findings are constrained by the spatially uneven distribution of these stations, and they might miss important information in regions where data are scarce. This issue can be addressed by using grid-based runoff simulations (Alfieri et al., 2013; Kumar et al., 2013; Niu et al., 2011; Samaniego et al., 2010; Sutanudjaja et al., 2018). Grid-based runoff simulations enable the employment of spatio-temporal event detection algorithms, which have been already successfully applied to analyse extreme events such as heatwaves, droughts and extreme events in the carbon cycle (Fang and Lu, 2020; Luo et al., 2022; Vogel et al., 2020; Zscheischler et al., 2013, 2014) and which can be used to study flood connectivity in space and time.

The occurrence and characteristics of floods have been observed to change over recent decades, potentially as a response to global warming, which affects flood generation processes (Berghuijs et al., 2019a; Blöschl et al., 2017, 2019; Jiang et al., 2022, 2024; Tarasova et al., 2023). For instance, flood magnitude and timing were revealed to change due to region-dependent

shifting patterns of snowmelt, rainfall and soil moisture (Berghuijs et al., 2019b; Blöschl et al., 2017, 2019). Similarly, changes in flood generation processes can influence the anomalies in the frequency of river floods, i.e., flood-rich or flood-poor periods (Blöschl et al., 2020; Lun et al., 2020; Tarasova et al., 2023). Regarding changes in flood extent, Berghuijs et al. (2019a) reported a general growth in the flood synchrony scale over Europe from 1960 to 2010. Kemter et al. (2020) further suggested a strong association between these changes in extent and shifts in flood magnitude. However, flood extent can potentially

be influenced not only by changes in runoff magnitude but also by changes in the spatial dependence of high runoff events. For example, Bevacqua et al. (2021) attributed the change in spatial extent of winter precipitation extremes to the change of precipitation magnitude and spatial dependence, and further quantified their relative contributions.

In this study, we first apply a spatio-temporal event detection algorithm to state-of-the-art runoff simulations over the last 70 years over Europe to identify and characterize large connected flood events. We compare our identified floods against an

independent flood impact database to evaluate our approach. We then quantify changes in key flood characteristics such as the spatial extent, and attribute changes to changes in the flood generation processes as well as changes in flood magnitude and dependence.

## 2 Data and methods

### 2.1 Data

We use daily routed runoff simulations over Europe from the hydrological model (mesoscale Hydrologic Model, mHM; Kumar et al., 2013; Samaniego et al., 2010) driven by observational data (E-OBS, version: 25.0e, Cornes et al., 2018) for the period 1951–2020. The mHM model routed runoff is available from the UFZ data portal (Rakovec et al., 2023). The spatial resolution of the routed runoff is 0.125°. The model setup is based on an earlier study (Rakovec et al., 2022) and has been recently extensively evaluated against multiple point ICOS observations across Europe (Pohl et al., 2023). Here, we used the

Hargreaves-Samani method (Hargreaves and Samani, 1985) to estimate daily potential evapotranspiration. For the spin-up of mHM, the model was firstly initialized in 1950-1959, and then the restart file of 1959 was read as an initial condition in 1940 run until 1949. This means that in total, 20 years were used to create steady-state conditions starting in 1950. A spatial mask is further applied to exclude catchments with headwater/contributing areas outside the E-OBS domain (Lehner et al., 2008).

To verify the robustness of our findings and to address model and forcing uncertainties, we use data from two additional grid-

ded runoff simulations: the Global Flood Awareness System (GloFAS; https://www.globalfloods.eu/, last access: 20 September 2023, Alfieri et al., 2013) and mHM simulations driven by the fifth generation of the European Center for Medium-Range Weather Forecasts (ECMWF) atmospheric reanalysis datasets (ERA5). GloFAS spans from 1979 to 2020, with a spatial resolution of 0.05°, while mHM simulation driven by ERA5 forcings extends from 1960 to 2020 with a resolution of 0.125°. Note that to compare the results, we only consider the periods overlapping with mHM simulations driven by E-OBS data.

We further use precipitation and temperature from E-OBS and snowmelt and soil moisture derived from mHM for flood event classification and trend attribution. To estimate population exposure to floods, we use version 4 of the Gridded Population of the World (GPWv4) at year 2000 (Center for International Earth Science Information Network - CIESIN - Columbia University,

2018). We aggregated the population data by summing up the population at a 2.5 arc-minute resolution within each 0.125° grid cell to match the resolution of the mHM simulations.

## 2.2 Model evaluation

To evaluate mHM model performance we use runoff stations from Global Runoff Data Centre (GRDC) dataset (https://www. bafg.de/GRDC, last access: 1 November 2021). Specifically, we first map GRDC runoff stations to corresponding mHM grid cells by identifying stations and grid cells whose catchment areas differ by less than 10%, these are 361 in total. The predictive accuracy of mHM simulations is accessed with the Nash–Sutcliffe efficiency (NSE) (Nash and Sutcliffe, 1970). Then, following Tarasova et al. (2023), we compare the simulated annual maximum runoff and peak timing with observations. In addition, we identify consecutive flood days at each grid cell (termed flood spells) and further evaluate mHM's capability to replicate the spatio-temporal organization of runoff extremes by comparing the number and trends of flood spells that constitute flood events.

In addition to the grid-based model evaluation, we further compare the identified set of spatio-temporal flood events (see Section 2.3) against an impact-based European flood record (Natural Hazards in Europe, HANZE, Paprotny et al., 2018), spanning from 1870 to 2016. Hereby, we compare the flood-affected regions provided by HANZE against the areas we have identified as flooded. An observed flood is regarded as detected in our dataset if the recorded flooding region overlaps with any identified floods within a ±3-day time window, and vice versa. It is important to note that HANZE includes only destructive floods with significant damage to people or property, while our identified floods encompass hydrological extremes, which may not necessarily result in damage, for instance in the case of little exposure and/or vulnerability. Note that the HANZE dataset is not used for formal validation of our flood events due to the large differences in the severity levels of floods between these two datasets. Instead, it serves more as a comparison to assess our flood detection algorithm's adequacy, particularly in capturing impactful floods, even with a relatively moderate detection threshold.

## 2.3 Identification and characterization of flood event

To identify the spatio-temporal evolution of flood events, we define flood days through the Peak over Threshold (POT) approach (Merz et al., 2016; Liu and Zhang, 2017). We consider a day as flood day when runoff surpasses the local 99th percentile based on the reference period of 1951–1980, and construct flood events by concatenating spatio-temporally connected flood days (Fang and Lu, 2020; Vogel et al., 2020; Zscheischler et al., 2013, 2014). To this end, we first create spatially connected flood patches at each time step by grouping neighboring flood days if they share an edge or vertex (at least one of 8 neighbors must be a flood day). We then calculate the overlap ratio between flood patches over two consecutive time steps. If the area ratios of the overlap region to both the previous and present patches are larger than 0.4, these two time steps are further combined to form a single event (Fang and Lu, 2020). Our results are not sensitive to the choice this overlap ratio.

Once the flood event database is established, we proceed to quantify their characteristics, including extent, duration, intensity and seasonality. Extent denotes the total area affected by the flood event; duration measures the time between the first start and last end dates; and intensity is computed by summing the routed runoff exceedance relative to the detection threshold (99th percentile) over all grid cells within a spatiotemporal event. Note that, to exclude non-riverine floods, our analysis focuses

exclusively on flood events involving at least one grid cell of a main stream, that is, a river with a catchment area larger than $1\,000\ \mathrm{km^2}$. The intensity calculation is limited to these main stream grid cells. Additionally, we define flood mean seasonality at each grid cell as follows: we first assign the first start date of flood events to each grid cell within the respective event and then average over all start dates in a grid cell using circular statistics (Hall and Blöschl, 2018). When assessing spatial patterns, we assign the characteristics of spatio-temporal flood events to each grid cell within the event.

### 2.4 Classification of flood events

To explore the relationship between flood extent and flood drivers, we categorize flood events into two primary types: snowmelt-driven and rainfall-driven flood events. To this end, we adapted the classification method from Tarasova et al. (2023) to better align with our event-based classification approach. Specifically, we first define the *contributing period* at each grid cell within the spatiotemporal flood event as the time window starting from six days before the onset of the flood event (at the considered grid cell) to the end of the event. Then the total amount of precipitation and snowmelt within the contributing period is calculated and compared. If the ratio of snowmelt amount to total rainfall surpasses 0.7 (Tarasova et al., 2023), the flood is categorized as snowmelt-driven, otherwise it is regarded as rainfall-driven. Using different thresholds does not affect our conclusions.

### 2.5 Attribution of changes in flood extent

In general, the occurrence of a compound event is shaped by the multivariate probability density function (pdf) associated with the variables describing the event (François and Vrac, 2023; Singh et al., 2021; Zscheischler and Seneviratne, 2017). According to copula theory, a multivariate pdf can be decomposed into the marginal pdfs and the copula describing the dependence between the individual variables (Sklar, 1973). Therefore, compound event changes can be shaped by changes in the marginal pdfs and changes in the dependence structure (Bevacqua et al., 2021; Zscheischler and Seneviratne, 2017). In our specific case, the marginal distributions describe the runoff at individual locations, while the dependence describes the dependence between the runoff at different locations. Quantifying the contribution to the compound event changes from marginal distribution and dependencies is common in compound event research as it provides insights into the origins of changes (François and Vrac, 2023; Bevacqua et al., 2021, 2020a).

Here, we quantify the change in flood extent as $100 \times (E_{pres} - E_{past})/E_{past}$, where $E_{past}$ and $E_{pres}$ denote the average flood extent over the past period (1951–1980) and present period (1991–2020), respectively. Following Bevacqua et al. (2021), we further quantify the contribution to changes in flood extent from changes in (1) runoff magnitude and (2) spatial dependence in the runoff field. Specifically, for each grid cell, we estimate (1) as: $100 \times (E_{magn} - E_{past})/E_{past}$, where $E_{magn}$ denotes the flood extent under the hypothetical condition that only the runoff magnitude changes without changes in dependence. $E_{magn}$ is derived from flood events in a spatiotemporal field of river runoff obtained via transforming data at each grid cell as $F_{R_{pres}}^{-1}(F_{R_{past}}(R_{past}))$, where $R_{past}$ and $F_{R_{past}}$ are routed runoff and its empirical cumulative distribution function (ECDF) in the past period, respectively; similarly, $F_{R_{pres}}$ is the ECDF of the present-day $R_{pres}$. Note that this corresponds to adjusting the ECDF of runoff over the past period based on the ECDF of the present period using quantile mapping (Cannon et al., 2015).

Based on this new hypothetical river runoff $(F_{R_{pres}}^{-1}(F_{R_{past}}(R_{past})))$ at each grid cell, we again identify spatio-temporally connected flood events, quantify their spatial extents, and assign the extents to each grid cell within the corresponding event to obtain the $E_{magn}$. Accordingly, (2) is estimated as $100 \times (E_{dep} - E_{past})/E_{past}$, where $E_{dep}$ denotes the flood extent under the hypothetical condition that only the dependence changes without changes in the runoff magnitude. To achieve this, we estimate flood extents $E_{dep}$ from the transformed runoff $F_{R_{past}}^{-1}(F_{R_{pres}}(R_{pres}))$.

We tested the significance of the difference between the two periods via Student's $t$-test using a significant level of 0.05. The overall work flow of this paper is presented in Figure 1.

## 3 Results

### 3.1 Model and flood detection evaluation

The mHM model generally performs satisfactory in simulating routed runoff, with a NSE over half of the GRDC stations exceeding 0.5 (Figure 2a). However, there are spatial variations in model performance. The mHM model achieves its highest performance in parts of western Europe, including Germany and England, where the NSE reaches 0.8. In contrast, the model performs relatively poorly in northern Europe, particularly in lake-rich southern Finland. Note that, since our primary goal is to quantify the spatial extent of flood events accurately, we focus more on the model's ability to capture the spatio-temporal organization of runoff extremes rather than the absolute value of daily runoff.

Hence, we further test the consistency between the number and trends of flood spells to evaluate model performance in reproducing the temporal structure of extreme streamflow. The mHM simulation shows largely consistent spatial patterns with GRDC stations for both the number and trends of flood spells, despite slightly overestimating trends in flood spells in northern Europe. The spatial correlation between the number and trends of simulated flood spells and the observed ones is 0.82 and 0.62, respectively. Additionally, the mHM model shows decent performance in simulating magnitude (correlation: 0.7) and timing of annual maximum runoff (Figure A1), which is comparable to other studies (Tarasova et al., 2023).

To further assess the suitability of our flood detection algorithm, we compare the identified flood events with the impact-based HANZE dataset (Figure A2). Among the four types of recorded flood events in HANZE (i.e., river flood, flash flood, coastal flood and compound flood), we only consider the first two types of flood in this study, as the mHM model cannot simulate tide or storm surge. Our dataset successfully detects 75% of river floods (349 out of 483), and 39% of flash floods (244 out of 623) over the shared period of 1951–2016. The lower detection rate of flash floods can be attributed to its definition, that is, river floods lasting less than 24 hours. Consequently, flash floods are very short with small spatial extent (Figure A2c, d), making them more challenging to detect. Also, many flash floods are located in coastal regions, which might add to the difficulty of detection (Figure A2b). Notably, our study only focuses on flood events involving grid cells with main streams, therefore potentially excluding certain small rivers.

Turning the comparison around, when focusing on the largest 100 detected spatio-temporal flood events, we find that 36 of those are documented in HANZE. Many of the largest floods detected in mHM simulations occur in northern Europe where population density is rather low, leading to insignificant or limited impacts on human society (Figure 3a). When flood intensity

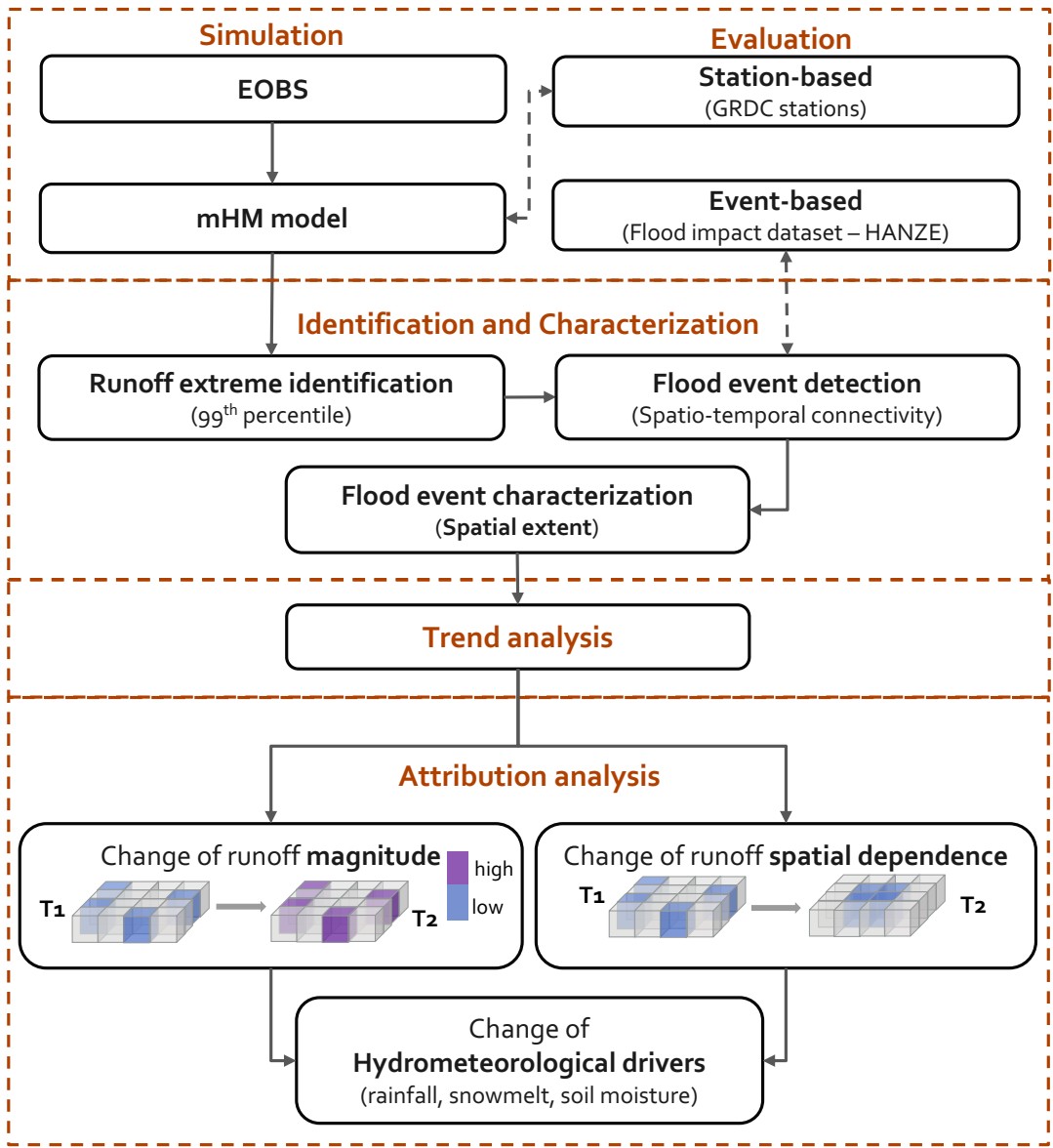

**Figure 1.** Main workflow of the study. The figure outlines the main analysis steps undertaken in this paper. In addition to the mHM model simulations driven by E-OBS data, we also analyze and compare the results obtained against mHM simulation driven by ERA5 data and the GloFas dataset.

is weighed by population exposure, most of the top 100 floods occur in populated regions like western and central Europe, and the detection rate increases to 50%. Overall, the consistency between our identified flood events and the HANZE dataset underscores the reliability of our data-driven flood database, rendering it suitable for further analysis.

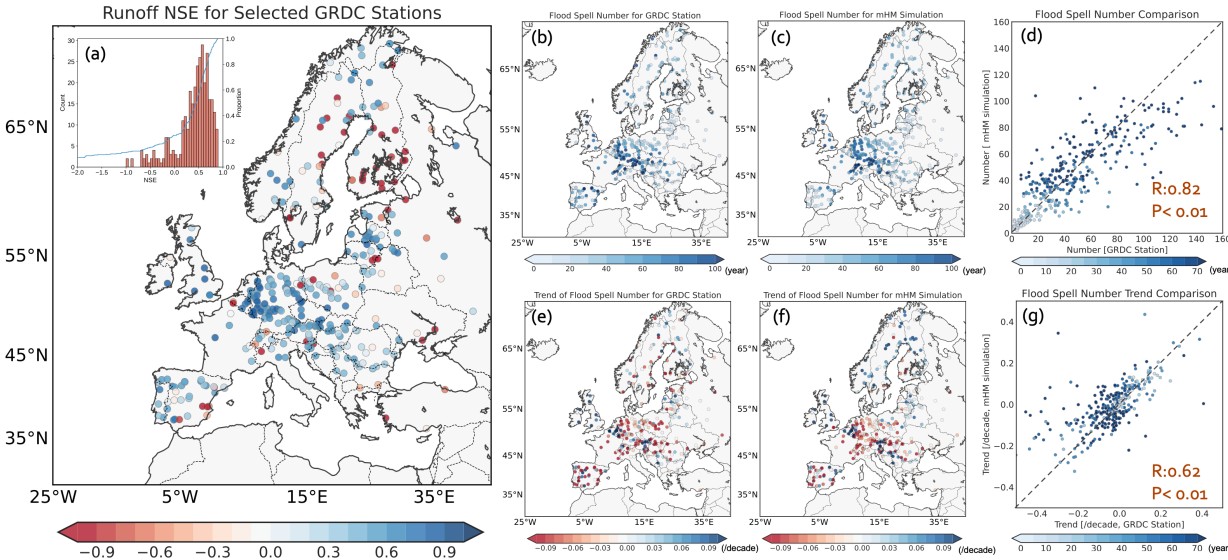

**Figure 2.** Evaluation of routed mHM runoff simulations. (a) Nash–Sutcliffe efficiency (NSE) between routed runoff from GRDC stations and corresponding mHM grid cells. Number of flood spells for (b) GRDC stations and (c) mHM grid cells. Trend of flood spells for (e) GRDC stations and (f) mHM grid cells. Scatterplot of the (d) number and (g) trend of flood spells between GRDC stations and mHM grid cells. Color bars in (e) and (f) denote the trend of spells (unit: spells/decade). Color bars in (d) and (g) denote the data length (unit: year) for selected GRDC stations.

## 3.2 Characteristics of large European floods

The mean flood extent over all detected floods is $6\,814$ km$^2$. We assign the characteristics of spatiotemporal flood events to each grid cell within the event to study (changes in) spatial patterns. Note that, as larger events cover many more grid cells, cell-based statistics are therefore weighted towards larger floods compared to event-based statistics. Floods in the plain regions (e.g., central and eastern Europe) are more widespread than those in mountainous regions like the Alps and Italy (Figure 4a). For example, the mean spatial extent of floods in eastern Europe, particularly along the Dnipro River, reaches up to $400\,000$ km$^2$, while floods in the Alps or mountainous region of Italy are much more localized, with an averaged spatial coverage smaller than $40\ 000$ km$^2$. Such spatial pattern of flood extent aligns closely with findings based on flood synchrony scale from a station-based perspective (Berghuijs et al., 2019a). Land-sea distribution may affect flood extent, for example, for regions like Italy, the surrounding sea may limit the potential for flood expansion. Since widespread floods typically last longer than more localised ones, the spatial distribution of flood duration mirrors the extent pattern (Figure 4b). As a result, and given the same frequency of total flood days across the study area (by definition, 1% of the study period), certain regions in western Europe, such as England and France, are prone to experiencing more frequent (Figure 4c) but shorter and smaller floods. In contrast, eastern and northern Europe are characterized by less frequent but longer and larger floods.

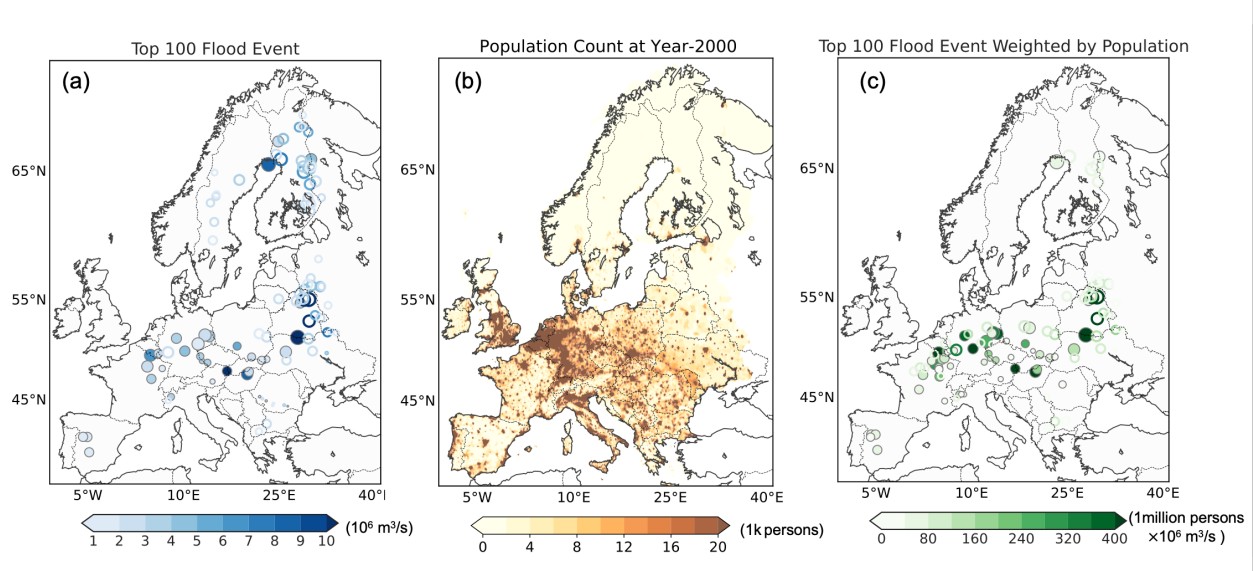

**Figure 3.** (a) Identified Top 100 flood events ranked by flood intensity (in $10^6 m^3 s^{-1}$) ; (b) European population count (in 1000 persons). (c) Top 100 flood events ranked by flood intensity weighted by population exposure (in 1 million persons $\times$ $10^6 m^3 s^{-1}$). The solid and hollow circles in (a) and (c) denote the flood events recorded and not recorded in the HANZE database, respectively. The size and color of the circles denote the spatial extent and intensity of flood events, respectively (see Section 2.3).

In the main manuscript, we report mean statistics, while median and 90th percentiles are provided in Appendix A (Figure A3). Mean statistics are strongly influenced by extreme values, given the skewed distribution of flood characteristics. All statistics depend somewhat on the employed threshold for flood detection (i.e., 99th percentile in this study).

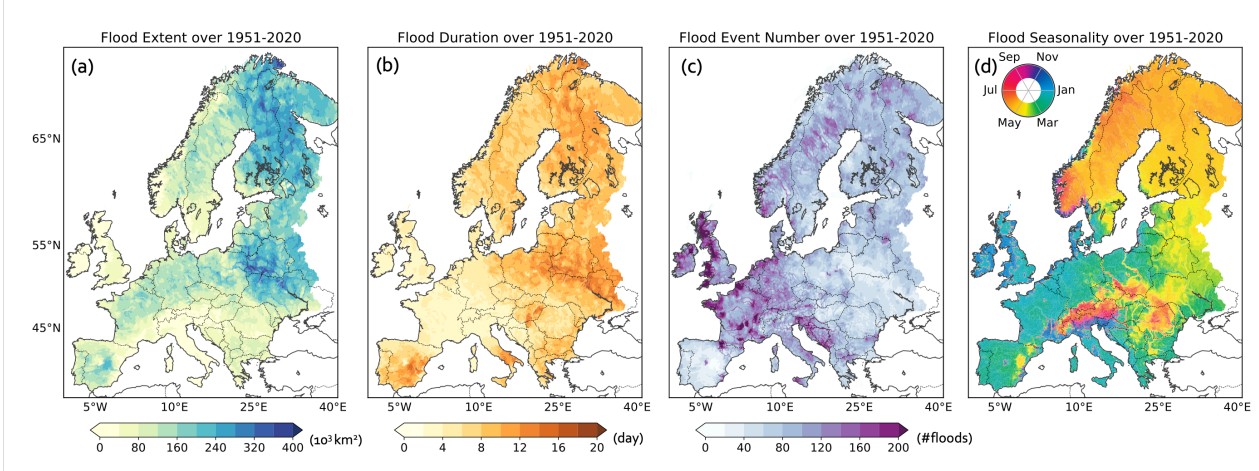

**Figure 4.** Flood event number and characteristics over 1951–2020. (a) Mean flood extent (in $10^3$ km$^2$). (b) Mean flood duration (in days). (c) Total number of flood events. (d) Flood seasonality, that is, averaged timing of the start date of flood events.

The spatial distribution of flood extents is not only influenced by topography but also by the flood generation process (Brunner and Fischer, 2022; Kemter et al., 2020). Inspired by the findings of Brunner and Fischer (2022), who highlight that snow-influenced floods exhibit stronger spatial connectedness than rainfall-driven ones, we categorize our flood events into snow-driven and rainfall-driven floods to investigate the role of the flood generation process in modulating flood extent. As expected and consistent with prior studies (Jiang et al., 2022, 2024; Kemter et al., 2020), snow-driven floods dominate northern and eastern Europe (Figure 5a) and typically occur in late spring (Figure 4d). Other high-snow fraction regions (Jiang et al., 2022), like the Alps and the Carpathian mountainous areas, are also dominated by snowmelt-induced floods, with a later flooding season in summer (Figure 4d). In contrast, rainfall-driven floods prevail in western and southern Europe (Figure 5c) and burst in winter (Figure 4d). A relatively clear dividing line between these two regions with different driving processes stretches from the Carpathian Mountains to the north of Poland (Figure 5a, c). Similar to Brunner and Fischer (2022), our findings indicate that snowmelt-driven floods tend to exhibit larger spatial expansion than precipitation-driven ones (Figure 5b, d), with mean extents of $8\,018$ km$^2$ and $6\,282$ km$^2$, respectively. Even within the same region, such as central and eastern Europe, the spatial extension of snowmelt-driven floods is nearly double that of rainfall-driven floods (Figure 5c, d). This difference is likely related to the greater spatial homogeneity of snowmelt, which results from relatively uniform warming in spring, as opposed to the more localized heavy precipitation events in winter (Brunner and Fischer, 2022). The presented results are robust under different thresholds for defining floods.

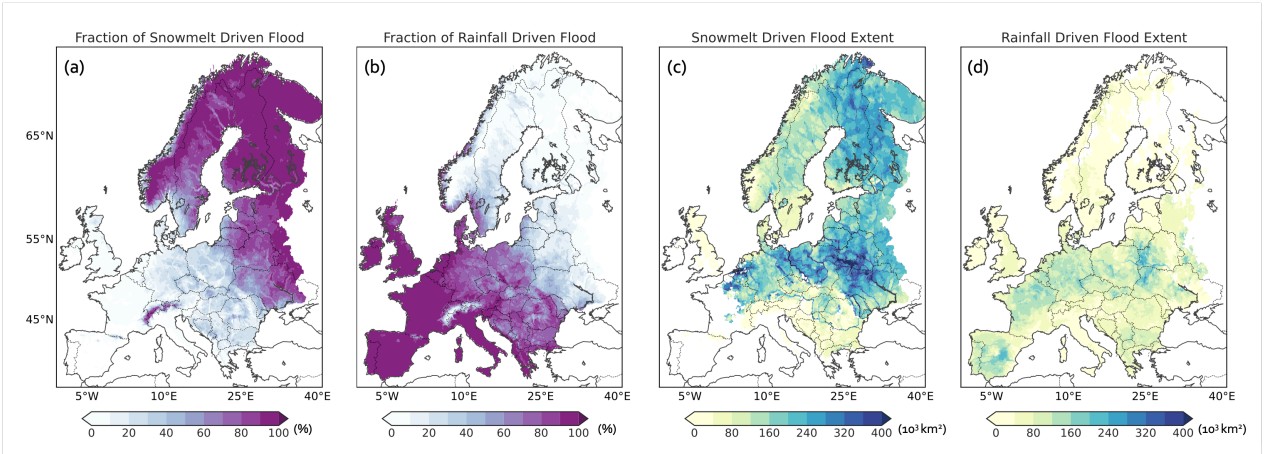

**Figure 5.** Fraction (%) of snowmelt-driven (a) and rainfall-driven(b) floods over the period 1951–2020. The spatial extent of snowmelt-driven (c) and rainfall-driven (d) flood events in $10^3$ km$^2$.

### 3.3 Increasing spatial extent of European floods

Averaged across Europe, both flood frequency and extent have increased over the past seven decades (Figure 6a). The extents have increased by 11.3% (Figure 6a), mostly related to the expansion of snowmelt-driven floods (Figure 6b). To assess spatial variations of these changes, we compare flood frequency and extent over two 30-year periods: 1951–1980 and 1991–2020.

We find that flood events increased in frequency by ~40% across the snowmelt-driven flood regions, i.e. in northern Europe (Figure 7a-c). Conversely, flood events decreased over parts of western (e.g., Germany), southern (e.g., Spain) and eastern Europe (e.g., Ukraine), except many coastal areas (Figure 7c). Large spatial variation is also observed in the change of flood extent (Figure 7d-f). Overall, 63.9% of the study area exhibits an increase in flood extent, and for 18% of the increasing regions changes are statistically significant (Figure 7f). Regions affected by an increase include parts of Norway and Finland in northern Europe, England, Denmark and Germany in western and central Europe, and Italy and Greece in southern Europe. In contrast, flood events in other regions tend to be more localised in the more recent decades, especially in parts of eastern Europe (e.g., Ukraine and Belarus) and southern Europe (e.g., Portugal and Spain).

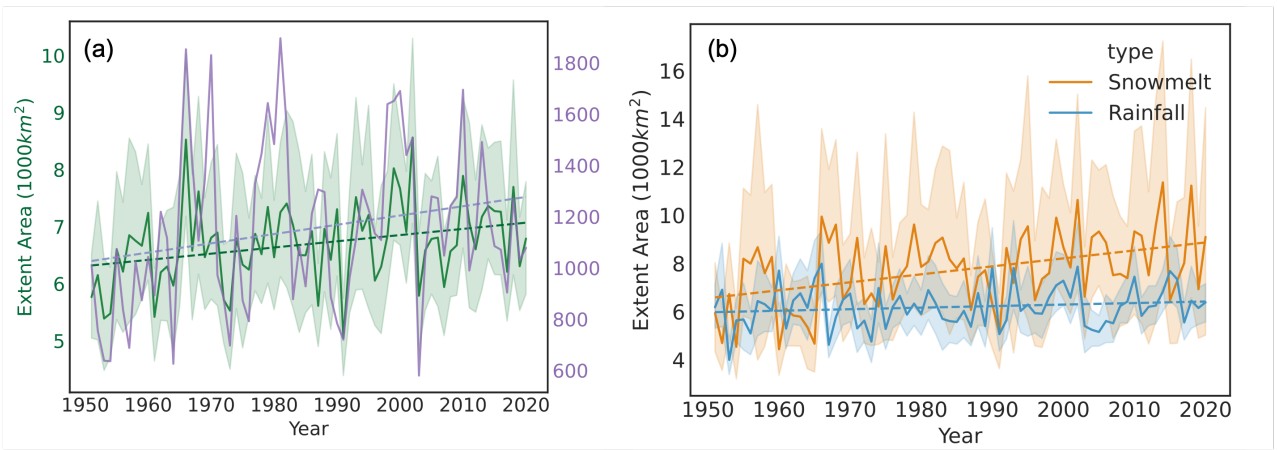

**Figure 6.** (a) Interannual variation of flood extent and flood frequency over 1951–2020. (b) Interannual variation of flood extent for snowmelt-driven and rainfall-driven floods over 1951–2020. The shading area in (a) and (b) denote the 90% confidence interval around the mean flood extent for each year.

To validate the robustness of our results, we assess uncertainties in changes arising from hydrological modelling and input forcing by incorporating two additional grid-based routed runoff datasets, namely GloFAS and mHM-ERA5. These datasets exhibit similar spatial patterns of average flood extent with mHM-EOBS, consistently showing more extensive flooding in plain regions compared to mountainous areas (Figure A4). We also observe general consistency in the shifts of flood extent between the models. Specifically, for flood extent changes based on mHM-ERA5 and mHM-EOBS over the periods 1961–1990 and 1991–2020, we identify an expansion in flood extent in parts of northern and western Europe, coupled with a reduction in eastern Europe and Spain. Similarly, when comparing the periods 1979–1999 and 2000–2020 for mHM-EOBS and GloFAS, the reduction in flood extent over Germany, France, Spain, and southern Sweden, along with the expansion in other regions are supported by both models (Figure A5). This overall consistency between results based on mHM-EOBS and two other datasets that use different models or input forcing underscores the robustness of our conclusions.

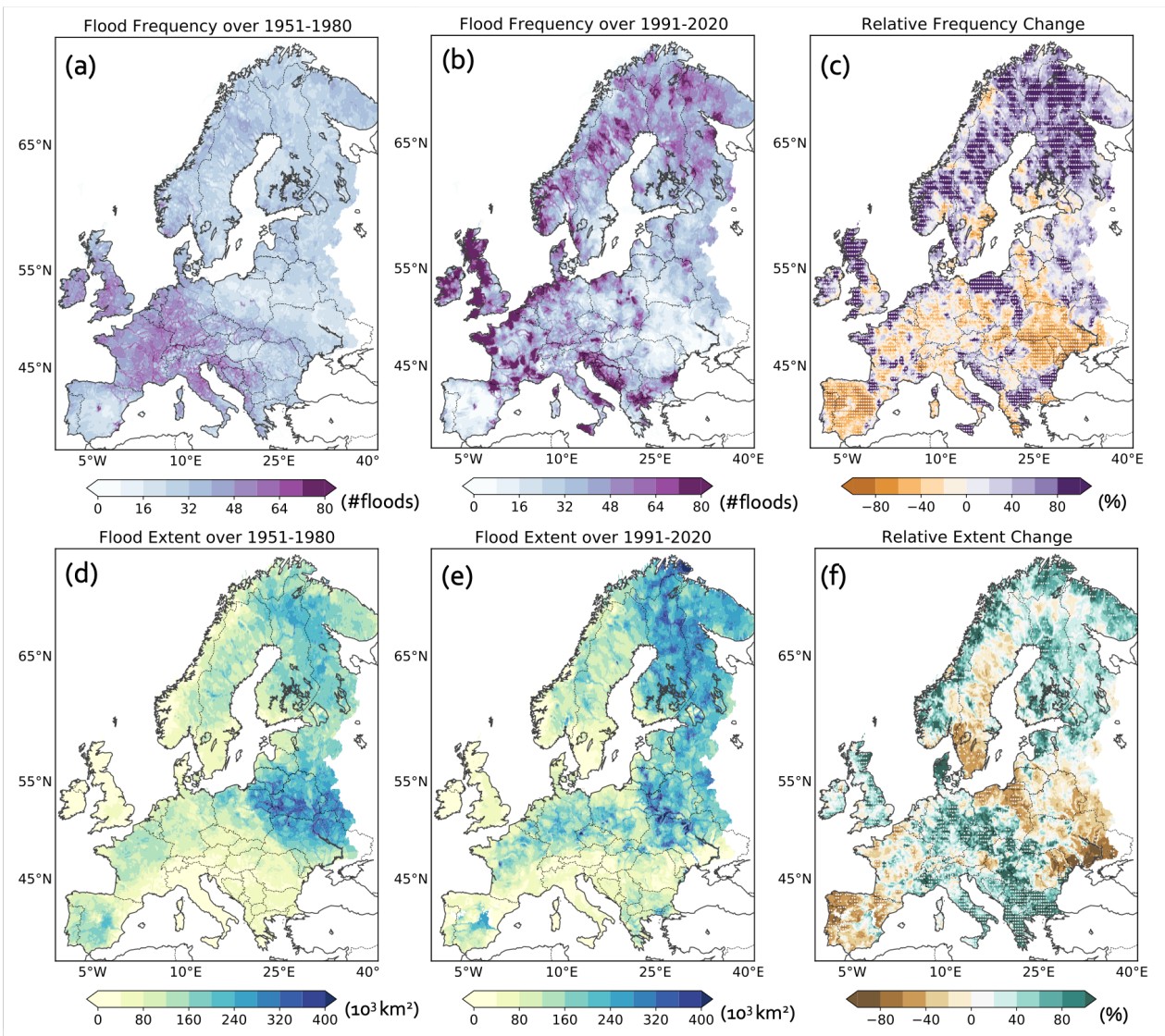

**Figure 7.** Total number of flood events over the period (a) 1951–1980 and (b) 1991–2020. Relative change (%) of the flood event number of 1991–2020 compared to 1951–1980. Spatial extent of flood events over the period (d) 1951–1980 and (e) 1991–2020. Relative change (%) of the flood extent of 1991–2020 compared to 1951–1980. The white dots in (c) and (f) denote whether changes are significant under Student t-test with significance level of 0.05.

## 3.4 Drivers of the changes in flood extent

The change of flood spatial scale has been reported to be highly correlated with a change in flood magnitude by a previous study (Kemter et al., 2020). Nonetheless, the alteration in the spatial dependence of the runoff fields, which also potentially contributes to the trend of flood extent, has received little attention. Therefore, building on previous studies (Bevacqua et al., 2021, 2020a; Zscheischler and Seneviratne, 2017), here we disentangle the contributions of both magnitude and spatial depen-

dence changes to the changes in flood spatial extent. Contributions from both sources exhibit significant spatial variability and dominate over distinct regions. In northern Europe, the expansion of flood extent emerges primarily from increasing runoff magnitudes (Figure 8a,d) due to intensified precipitation and snowmelt (Figure 9a, b). In contrast, the change in runoff spatial dependence contributes little, even slightly hindering the increase of flood scale (Figure 8b, d). It is worth noting an exception in southern Sweden where changes in runoff spatial organization (i.e., dependence) lead to a reduction in flood extent.

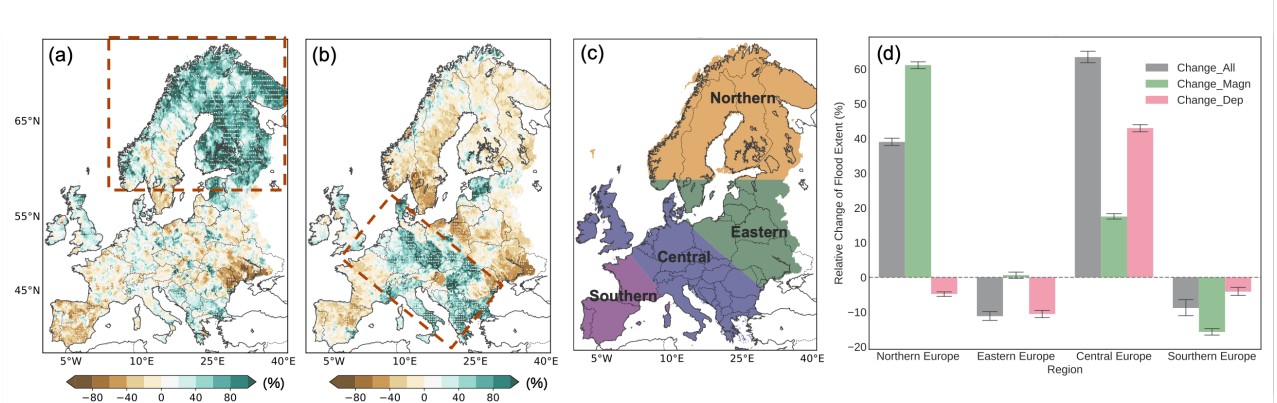

**Figure 8.** Relative change (%) in flood extent caused by the change of (a) runoff magnitude and (b) spatial dependence between the two periods 1951–1980 and 1991-2020. The red boxes in (a) and (b) denote regions where the change of runoff magnitude and spatial dependence significantly enlarges the flood extent, respectively. (c) Four sub-regions of the study area, losely based on the attribution results from (a) and (b). (d) Relative change in flood extent between 1951–1980 and 1991–2020 ("Change All"), and its decompositions over the four sub-regions defined in (c). "Change Magn" and "Change Dep" refer to change caused by runoff magnitude and runoff spatial dependence, respectively. Bars show mean flood extents over four defined regions in (c) and uncertainty estimators denote the 99% confidence interval around the regional mean flood extents.

In eastern European regions like Ukraine, Belarus, Lithuania, and northern Poland, changes in both runoff magnitude and its spatial dependence collectively contribute to a decrease in flood spatial extent, with the latter playing a more prominent role (Figure 8b, d). We suppose that the reduction in snowmelt in areas where snowmelt is the main floods' driver (Figure 9a) not only decreases runoff magnitude but possibly shrinks the extent of flood events due to a smaller synchronization scale in snowmelt. The combined effect of the change of runoff magnitude and spatial dependence is also evident in the region dom-

inated by rainfall-driven floods (see Figure 5c). Particularly in England, changes in runoff magnitude and spatial dependence together increase flood extent. In contrast, both drivers strongly reduce flood extent over Southern France, Portugal and Spain (Figure 8a, b). Overall, changes in soil moisture seem to play a relatively minor role in the detected changes in flood extent compared to rainfall and snowmelt (Figure 9c). For instance, in Germany and France, although soil moisture decreases, flood extent increases due to more extensive heavy rainfall, highlighting the dominant role of rainfall compared to soil moisture.

However, its impact can be substantial at the regional scale. For example, in the northern UK, a strong increase in soil moisture contributes to the increase in flood extent; while in northern Iberia, the decrease in soil moisture is associated with a reduction in flood extent. These findings align with those of Blöschl et al. (2019) and Bertola et al. (2021), who revealed a similar role

of soil moisture in influencing flood magnitude in these regions. Furthermore, in the Mediterranean Sea region, a decrease in flood extent is partly attributed to the decrease in soil moisture, generally aligning with the findings of Tarasova et al. (2023).

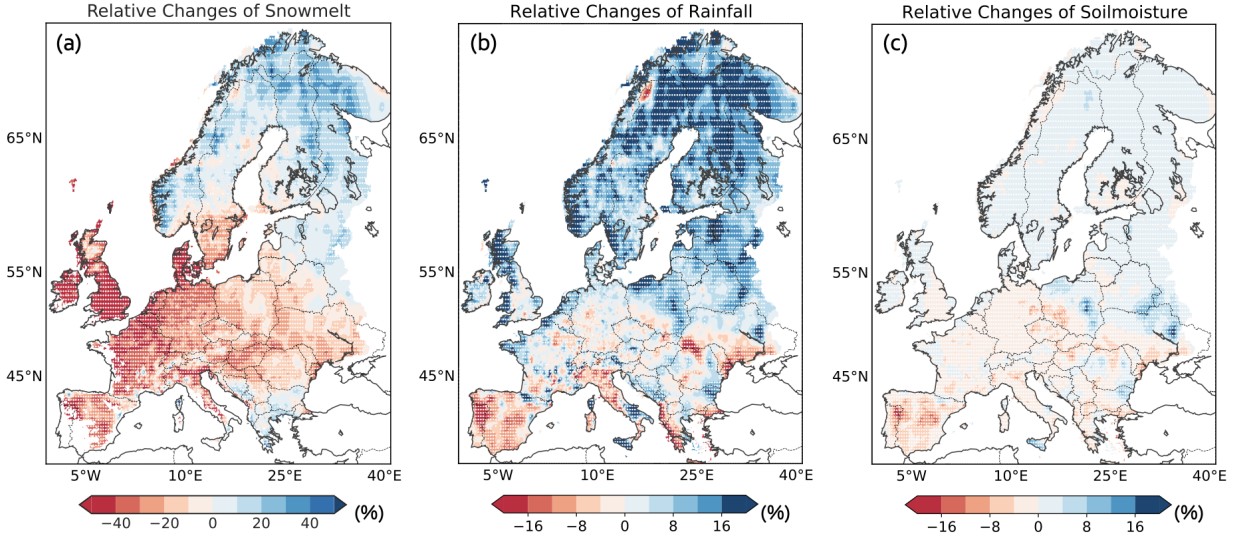

**Figure 9.** Relative change of (a) snowmelt, (b) rainfall and (c) soil moisture of all days over the period 1991–2020 compared to 1951–1980. White dots denote whether changes are significant under the Student t-test with a significance level of 0.05. In (a), only the grids with over 30 days of snowmelt larger than 2 mm are considered.

In contrast to the other regions, increases in runoff dependence increase flood extents by 40% to 80% across western, central and southern Europe, notably affecting Germany, southern Poland, Italy and Greece (Figure 8b, d), whereas runoff magnitude changes play a minor role in these areas (Figure 8a, d). This implies that, despite the negligible change in runoff magnitude, runoff extremes clustered in space more in the last three decades compared to 1951–1980, resulting in the observed expansion of flood events. As rainfall controls floods in these regions (see Figure 5c), this enhanced clustering may stem from changes in
the spatial organization of heavy rainfall. We therefore further assess changes in heavy rainfall extent over these two 30-year periods.

Heavy rainfall events are identified similarly to flood events but using the 80th percentile of wet days (precipitation $> 1$ mm) as a threshold to include potentially contributing heavy but not only extreme rainfall. Moreover, we retain only the five most extensive spatio-temporally connected rainfall events for each year to improve alignment with flood occurrences. The spatial
extent of heavy rainfall events exhibits large spatial variability across Europe (Figure 10a, b). Central and eastern Europe, characterized by lower altitudes, experience the broadest heavy rainfall extent, while the least extensive rainfall occurs in the mountainous regions of southern and northern Europe. This spatial pattern aligns with a similar pattern over the whole Northern Hemisphere found by Bevacqua et al. (2021), who posited that precipitation tends to be more widespread in flat terrains because of less topographic obstruction. Nevertheless, note that since our analysis exclusively focuses on terrestrial rainfall events, the
land-sea distribution also influences our spatial patterns.

When comparing the two periods 1991–2020 and 1951–1980, we find an overall increase in heavy rainfall extents in most areas, except for parts of France, Spain, England and Ireland (Figure 10). In regions dominated by rainfall-driven floods (Figure 5b), flood and rainfall extent are correlated (Figure A6). This suggests that the change in runoff spatial dependence might be attributed to the change in heavy rainfall extent in these regions. Specifically, in regions where the increase in spatial dependence of runoff significantly increases the flood extent (as denoted by the red box in Figure 8b, e.g., Germany and Greece), more widespread heavy rainfall results in more spatially connected runoff extremes, leading to spatially more extensive floods. Furthermore, consistent with the flood extent attribution results, the rise in heavy rainfall extent is also primarily driven by an increased spatial dependence of precipitation in the above regions (not shown). Conversely, in regions where the extent of heavy rainfall has substantially decreased, such as in southern France and Spain, the resulting reduction in spatial dependence of runoff leads to more localized floods. A summary of these attribution results is provided in Figure 11.

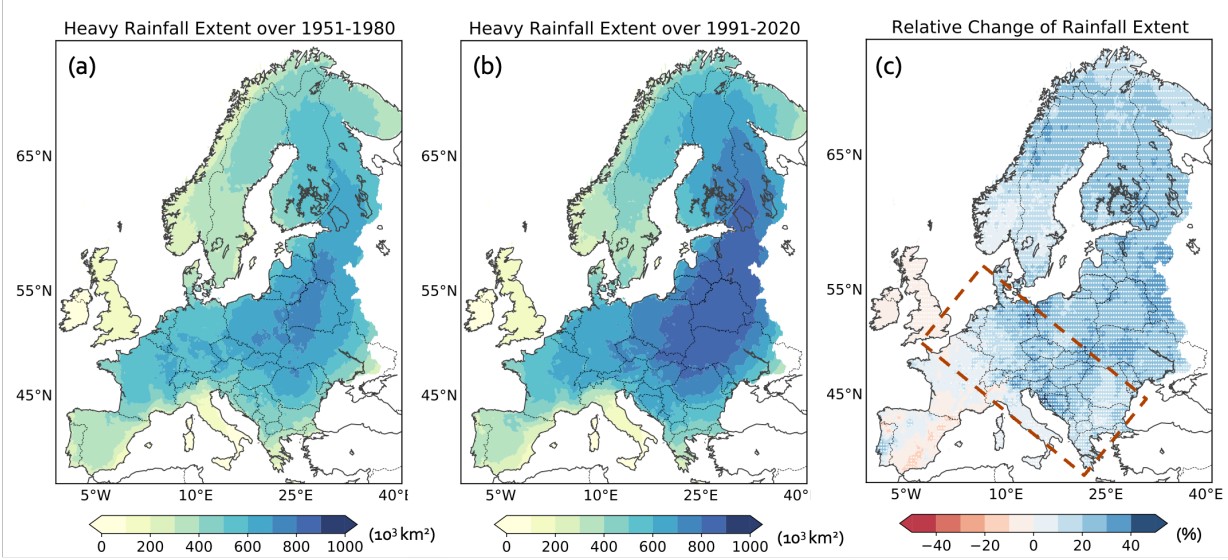

**Figure 10.** Mean spatial extent of heavy rainfall over the period (a) 1951–1980 and (b) 1991–2020 (unit: 1 000 km$^2$). Relative change (%) between these two periods (c). The red box is the same as in Figure 8b.

## 3.5 Discussion

Based on routed runoff simulations, we find that flood extent in Europe is influenced by both topography and the flood generation process. Generally, flood tends to be more widespread in plain than mountainous regions, with snow-driven floods being more widespread than rainfall-driven ones. We reveal a significant increase in flood extent over Europe over the last 70 years, with notable spatial variations. Specifically, the significant expansion of snowmelt-driven floods in northern Europe is primarily attributed to increased runoff magnitude, possibly due to intensified rainfall and snowmelt. In contrast, a reduction in snowmelt contributes to decreased flood extent in eastern Europe. In regions dominated by rainfall-driven floods, the increases in rainfall extent over eastern and southeastern Europe result in more spatially connected runoff extremes, leading to more extensive

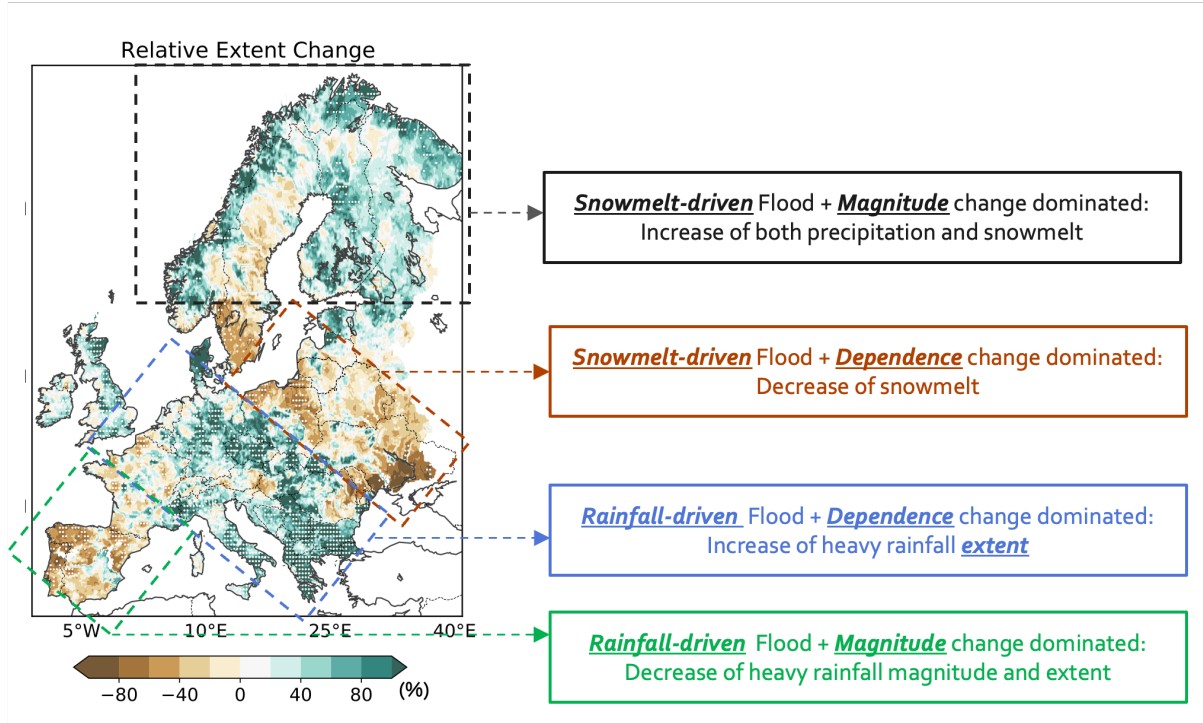

**Figure 11.** Schematic summarizing the attribution results of the change in flood extent to changes in runoff spatial dependence and magnitude under different flood generation processes. This figure of the "Relative Extent Change" is the same as Figure 7f.

floods. The decrease in heavy rainfall magnitude and extent result in reduced flood coverage over parts of western and southern Europe, including France, Portugal and Spain. While it is possible that these changes are driven by anthropogenic climate change, the observed large spatial variability in the changes in flood extent and associated changes in flooding processes could also be a result of internal climate variability (Bevacqua et al., 2023).

Our results on the spatial patterns and trends in flood extents align well with existing literature (Berghuijs et al., 2019a; Kemter et al., 2020), despite different studies employing different flood detection methods (peak over threshold vs annual maxima) and datasets (grid-based simulation vs station observations). However, some discrepancies do exist. For instance, in contrast to the findings of Jiang et al. (2022) and Tarasova et al. (2023), which suggest a significant decrease in snowmelt-induced flood events (defined by annual maxima) in northern Europe over the last 5–7 decades, our results indicate an overall increase in both their frequency and extent. We attribute these differences primarily to differences in the detection methods. In particular, in contrast to earlier studies, we analyse large spatio-temporal floods. We find that the increase in northern European floods is primarily driven by increases in rainfall and snowmelt magnitude (Figure. 9). This is supported by the finding that when we keep runoff magnitude to past levels (1951–1980) and only shift the spatial dependence to the present level (1991–2020), we observe a decreasing effect in snow-melt-driven flood extent stemming from changes in river runoff spatial dependence (Figure. 8).

The assessment of changes in flood extent across central and southeastern Europe (blue box in Figure 11), reveals seemingly contradictory results during different periods (see Figure 7f and Figure A5f), particularly in Germany. Specifically, flood extent significantly increases during 1991–2020 compared to 1951–1980, while a small reduction is found when comparing the period 2000–2020 with 1979–1999. This implies that the expansion of flood events has not continuously occurred over the last seven decades but is more likely a result of decadal variability (Bevacqua et al., 2023). This is supported by Tarasova et al. (2023), who identify a short but pronounced flood-poor period in the Atlantic region during 1970–1980, followed by a flood-rich period from the late 1980s to the mid-2000s. Hence, the absence of a significant shift in flood extent over two periods (2000–2020 vs. 1979–1999) which are both within the flood-rich period (1980s to 2000s), is not surprising. The substantial expansion of flood events during the flood-rich period compared to the flood-poor period also aligns with expectations. Furthermore, Merz et al. (2018) also reports a recent tendency of numerous catchments experiencing annual peak runoff simultaneously, indicating a stronger spatial coherence in flooding over Germany. Besides, the substantial increase in the 90th percentile of flood extent (Figure A3) suggests more frequent widespread floods in recent decades in Germany and southeastern Europe (e.g., Greece).

A formal attribution of changes in flood extent to anthropogenic climate change is not the focus of this study. Nevertheless, insights into its role can be partly inferred from the changes in flood drivers. For instance, the warming climate induces both an increase and decrease in snowmelt in northern and eastern Europe, contributing to the expansion and reduction of snow-driven flood extent over these two regions, respectively. This pattern aligns with findings by Stahl et al. (2012), who also highlight inconsistent trends in annual runoff in snow-dominant regions. In regions dominated by rainfall-driven floods, a general increase in extreme winter precipitation due to the increasing atmospheric water-holding capacity of warmer air (Blöschl et al., 2019; Stahl et al., 2012; Zolina, 2012), is found to be responsible for the expansion of flood events over western and central Europe (e.g., England, Germany). Conversely, in parts of southern Europe, decreasing or constant precipitation (Bevacqua et al., 2020b) combined with increased evapotranspiration due to higher temperatures, lead to a decrease in flood magnitude (Archer and Caldeira, 2008; Blöschl et al., 2019; Stahl et al., 2010) and flood extent. These hypotheses are in line with Gudmundsson et al. (2021) who attributed an observed overall increase in river runoff in northern Europe, and decrease in southern Europe to anthropogenic climate change.

When investigating the drivers of changes in flood extent, here we primarily focus on the changes in snowmelt and rainfall. In particular, we mainly attribute changes in rainfall-induced flood regions to shifts in precipitation extent. Another relevant driver, soil moisture, is not extensively explored here because of its relatively much smaller trend compared to the other drivers (Figure 9c). Nevertheless, building on Tarasova et al. (2023) who report an increase in the frequency of "rain on wet" and "rain on dry" floods in the Atlantic and Mediterranean regions, we explored representative cases in Germany, England, Italy and Greece to examine the associations between soil moisture, rainfall and their contributions to shifts in flood extent (Figure A7). Our findings largely align with Tarasova et al. (2023). Specifically, in the selected Atlantic region (i.e., England and Germany), the observed increase in flood extent is associated higher precipitation and wetter soil conditions. Conversely, in the Mediterranean region (i.e., Italy and Greece), despite of drier soil moisture (Figure A7c, d), more widespread heavy rainfall events still lead to an increase in flood extent.

The reliability of our results heavily relies on the model's performance in simulating the spatiotemporal organization of runoff extremes. As noted in the model evaluation section (Section 3.1), the mHM model generally performs well in simulating runoff extremes (e.g., annual maxima) and their spatio-temporal connectivity. Furthermore, the resulting flood event database aligns closely with an independent impact-based flood record database by capturing 75% of recorded impactful river floods, providing a robust foundation for characterizing floods and attributing changes. However, it is important to recognize the spatial variability in the model's performance. For instance, mHM exhibits limitations in simulating runoff over lake-dominated regions, leading to lower model performance in southern Finland. Additionally, there is an overestimation of flood spell trends in parts of northern Europe, potentially leading to an overestimation of flood extent trends in those areas. These spatial discrepancies should be considered when applying our findings to regions where the model's reliability may be lower. Furthermore, we note that no information on dam and reservoir constructions, which might change flood dynamics over time, are included in the model setup. We are unaware of a database that would harmonise such information across the entire European domain over time. Possibly even more crucial, bathymetric data, which would be needed for sedimentation processes, is not available.

Uncertainties in the simulations do not only stem from the hydrological model, but also from the weather data input, here the E-OBS dataset. While E-OBS is a widely employed observation-based meteorological dataset, it has some limitations. In particular, the heterogeneity in station coverage across different regions can introduce spatial variations in temperature, precipitation, and, subsequently, runoff (Cornes et al., 2018). This issue is particularly evident in areas like the Mediterranean and eastern Europe, where lower station density may exaggerate flood extent values due to increased spatial smoothness (Rivoire et al., 2021), However, this issue is mitigated by focusing on regions where E-OBS reliability is not clearly compromised (see Section 2.1). Furthermore, the temporal evolution of the station density, with an increase from the 1950s to the 1990s and a subsequent decrease, can affect the temporal variability of runoff data. This, in turn, can potentially affect the accuracy of the trend analysis regarding flood extent. To address these uncertainties, we incorporated two additional routed runoff datasets, namely GloFAS and mHM-ERA5, to validate the robustness of our results. The general consistency observed in the spatial patterns of flood extent and its relative changes during the two selected periods provides confidence in our results.

We also acknowledge certain limitations inherent in our flood detection and classification algorithms. In our flood classification process, we limit our analysis to rainfall and snowmelt that occur directly within the spatial area, that is, the same pixels of the flood events, while the precipitation and snowmelt falling outside the event but within the corresponding hydrological catchment are not considered. Nevertheless, the high coherence between flooding across grid cells within specific catchments and the outlet grid cells, largely addresses this issue. Specifically, as shown in Figure A8, if an outlet grid cell is experiencing flooding, there is a high probability that grid cells within the corresponding catchment also experience flooding. This suggests that even though we do not consider the entire contributing catchment for each grid cell (which is often infeasible), our event-based results remain plausible. Furthermore, the general alignment of our classification results with prior studies (Jiang et al., 2022; Kemter et al., 2020) underscores the robustness of our classification algorithm.

## 4    Conclusions

Previous research has predominantly examined flood events from the standpoint of individual stations or single regions. Despite the dangers associated with simultaneous floods across multiple rivers and regions, the quantification of the floods' spatial extent, a factor that largely shapes the total impact of flood events, has received limited attention. Therefore, this study employs a three-dimensional detection algorithm and routed runoff simulations derived from a state-of-the-art hydrological model to identify the spatio-temporally connected flood events, quantify their spatial extents along with other relevant characteristics, and inspect the drivers of changes. Our findings reveal that floods are more widespread in low-lying regions, such as parts of western and eastern Europe, than in high mountainous regions like the Alps. Furthermore, snowmelt-driven floods are more widespread than rainfall-induced floods, which can be attributed to the larger spatial coherence of snowmelt in the spring season across northern Europe in contrast to more localized, intense winter precipitation in the Atlantic, North Sea, and Mediterranean regions.

Averaged across Europe, we identify an increase of 11.3% in flood extent over the last 70 years. However, the trend is highly spatially variable. While there is a significant increase in flood extent in northern Europe, the North Sea and Mediterranean regions, there is a significant decrease in eastern Europe, Portugal and Spain. We attribute the increase in northern Europe to increased precipitation and snowmelt. Furthermore, by decomposing the change in flood extent into changes in runoff magnitude and spatial dependence, we find that an increase in runoff magnitude increases the flood extent in northern Europe, whereas a decrease in runoff magnitude reduces the flood extent in parts of southern Europe such as Portugal and Spain. In contrast, an increase in the spatial dependence of runoff extremes, likely a response from more widespread heavy rainfall, increases flood extents in parts of western Europe (including Germany), central Europe (including Czech Republic), and southern Europe (including Italy and Greece).

Over the past seven decades, the substantial expansion of flood extents in Europe has posed growing environmental and societal challenges. Concurrent floods across multiple regions and basins carry the potential for more extensive and severe impacts compared to localized floods. This calls for enhanced collaboration among multi-regional and multi-national government agencies responsible for water resource management and flood control to mitigate the compounding impacts of large river floods (Jongman et al., 2014). Under a continuously warming climate, extreme rainfall will occur more frequently and cover wider spatial scales over large parts of Europe (Bevacqua et al., 2021), which may lead to an increase in the occurrence of widespread and devastating floods. We have shown that flood extent is influenced not only by runoff magnitude but also by its spatio-temporal organization. Hence, even regions with relatively stable runoff or precipitation magnitude may experience changes in flood extent when the spatial dependence of runoff extremes changes.

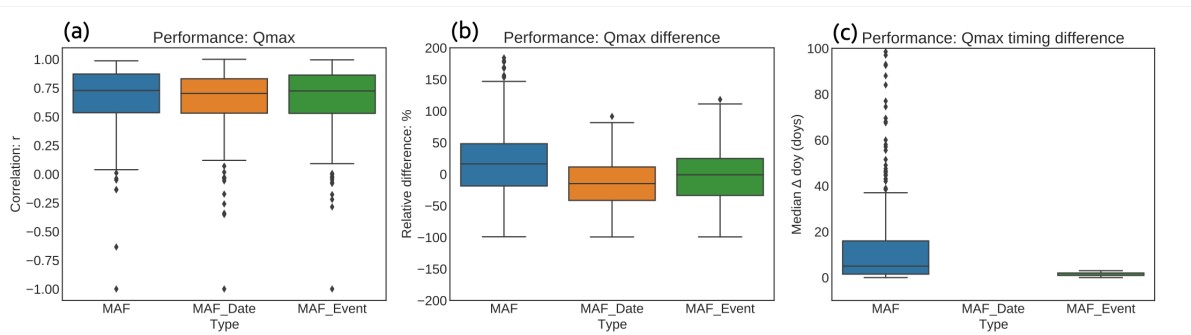

**Figure A1.** Model performance in terms of timing and magnitude of annual maximum floods under three comparison scenarios (Tarasova et al., 2023). Specifically, "MAF" compares observed and simulated maximum annual floods in terms of their discharge and day of occurrence; "MAF date" compares the observed and simulated discharge on the exact date of the observed annual floods; "MAF event" compares observed discharge and date of maximum annual flood to the peak of corresponding runoff event. More details about these three cases can be found in Tarasova et al. (2023). (a) Pearson correlation; (b) relative difference between the magnitude of simulated annual maximum runoff with observations from GRDC stations; (c) relative difference in peak timing between simulated annual maximum runoff with observations from GRDC stations.

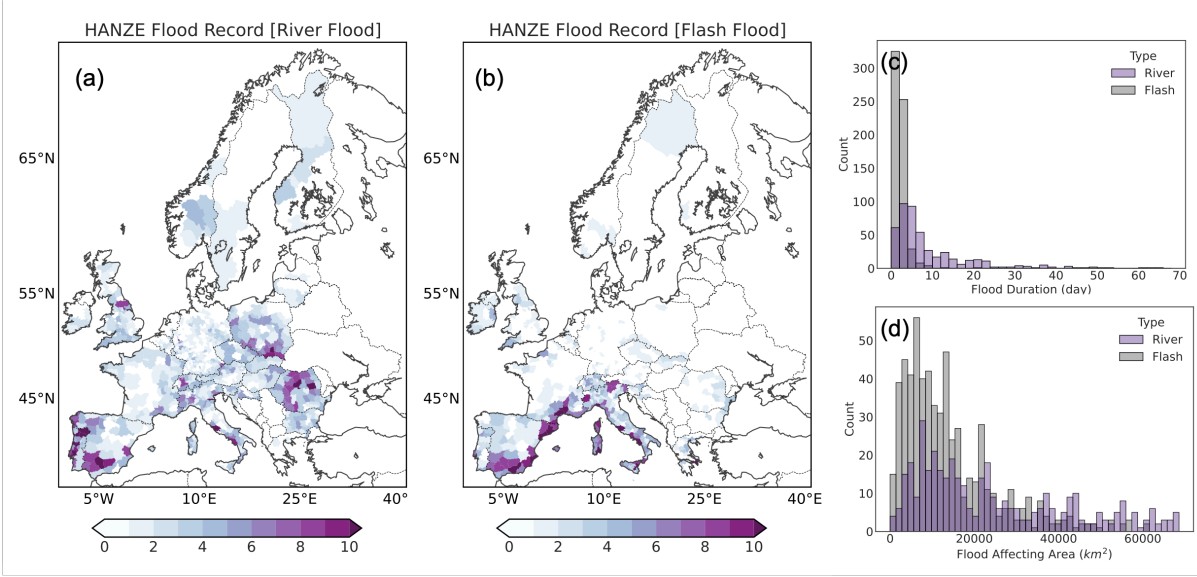

**Figure A2.** Overview of the HANZE flood impact database. Spatial distribution of the number of (a) river floods and (b) flash floods during the period of 1951–2016 (NUTS3 regions). (c) Duration in days and (d) affected area in km$^2$ for river floods and flash floods.

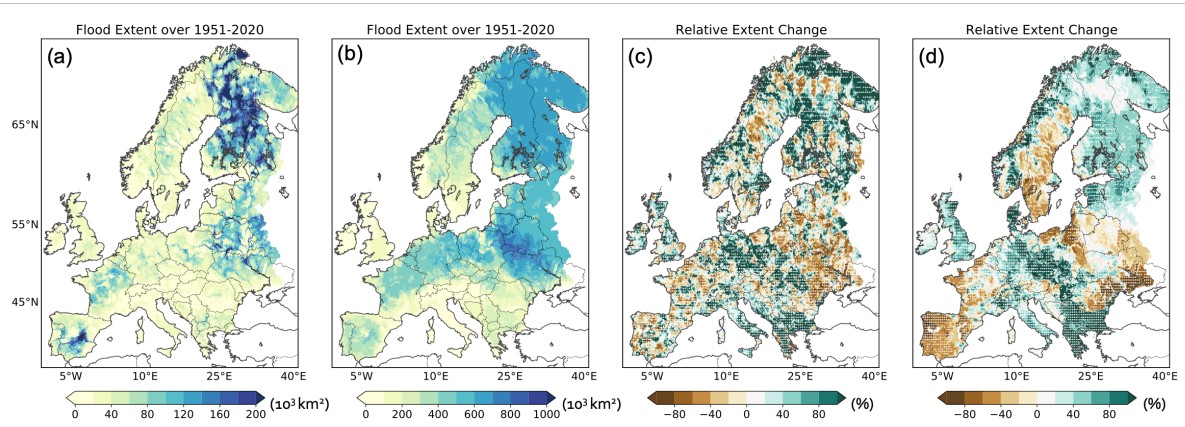

**Figure A3.** Spatial extent (a, b) and changes in spatial extent (c, d) of flood events over the period 1951–2020. (a, c) based on the median, (b, d) based on the 90th percentile. Changes denote the relative difference (%) between 1991–2020 and 1951–1980. The white dots in (c) and (d) denote whether changes are significant based on the bootstrapping method with a significance level of 0.05.

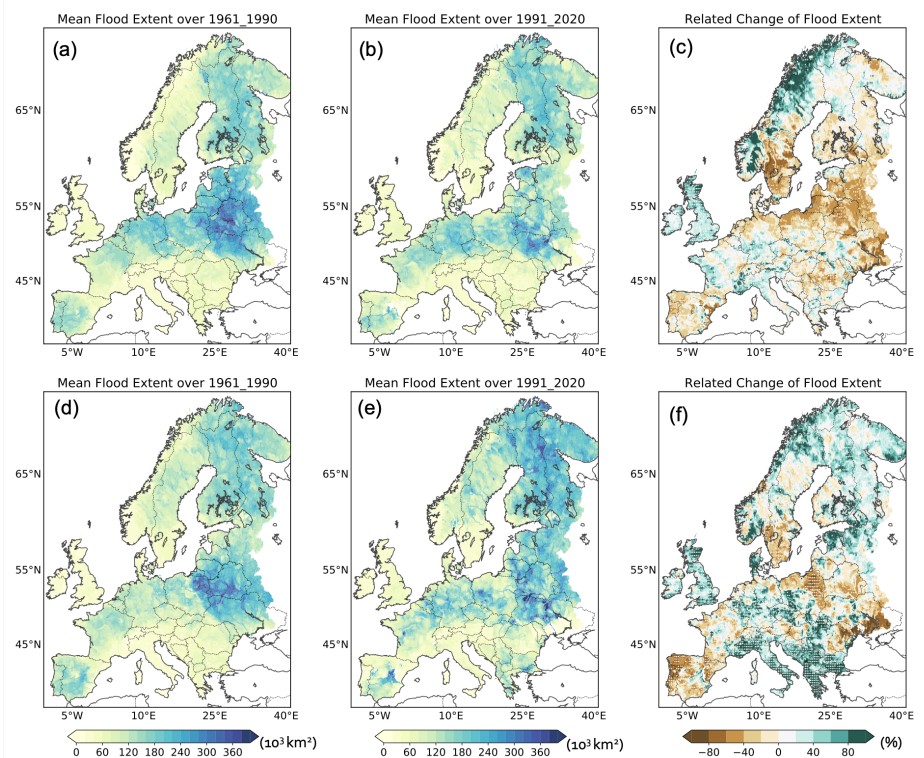

**Figure A4.** Mean spatial extent of flood event derived from (a-c) mHM-EOBS and (d-f) mHM-ERA5 datasets during (a,d) 1961–1990 and (b, e) 1991–2020 (unit: $10^3$ km$^2$). (c, f) relative change (%) in flood extent over these two periods.

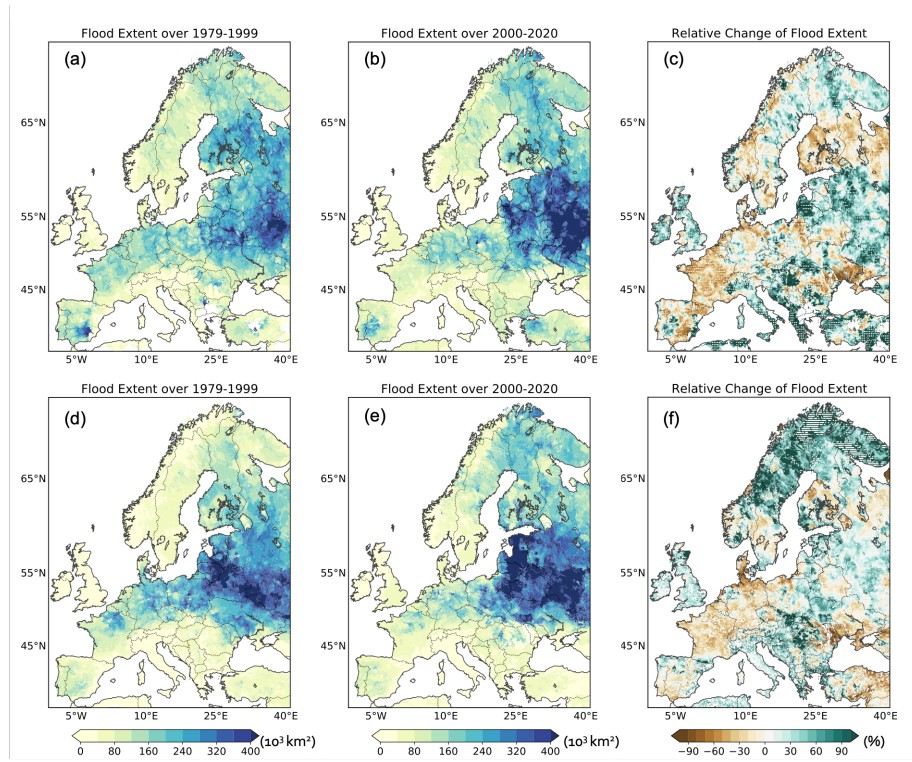

**Figure A5.** Mean spatial extent of flood event derived from (a-c) mHM-EOBS and (d-f) GloFAS datasets over during (a, d) 1979–1999 and (b,e) 2000–2020 (unit: $10^3$ km$^2$). (c, f) relative change (%) in flood extent over these two periods.

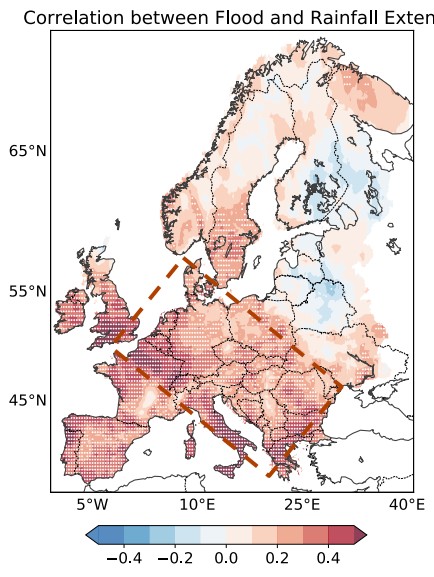

**Figure A6.** Spearman correlation between the annual mean spatial extent of flood and heavy rainfall extent over the period of 1951–2020. The white dots denote the significant correlated grids with $p < 0.05$. The red box region is the same as in Figure 8b.

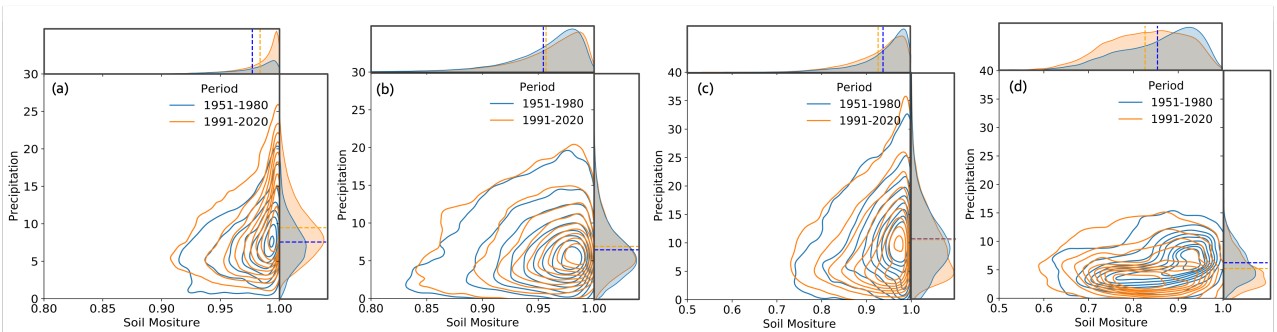

**Figure A7.** 2D Kernel density estimate (KDE) plot of soil moisture (unit: mm$^3$/mm$^3$) and precipitation (unit: mm day$^{-1}$) for identified events over four regions: (a) England; (b) Germany; (c) Italy; (d) Greece based on period 1951–1980 and 1991–2020. The marginal distributions show 1d-KDE for soil moisture and precipitation over these two periods, respectively.

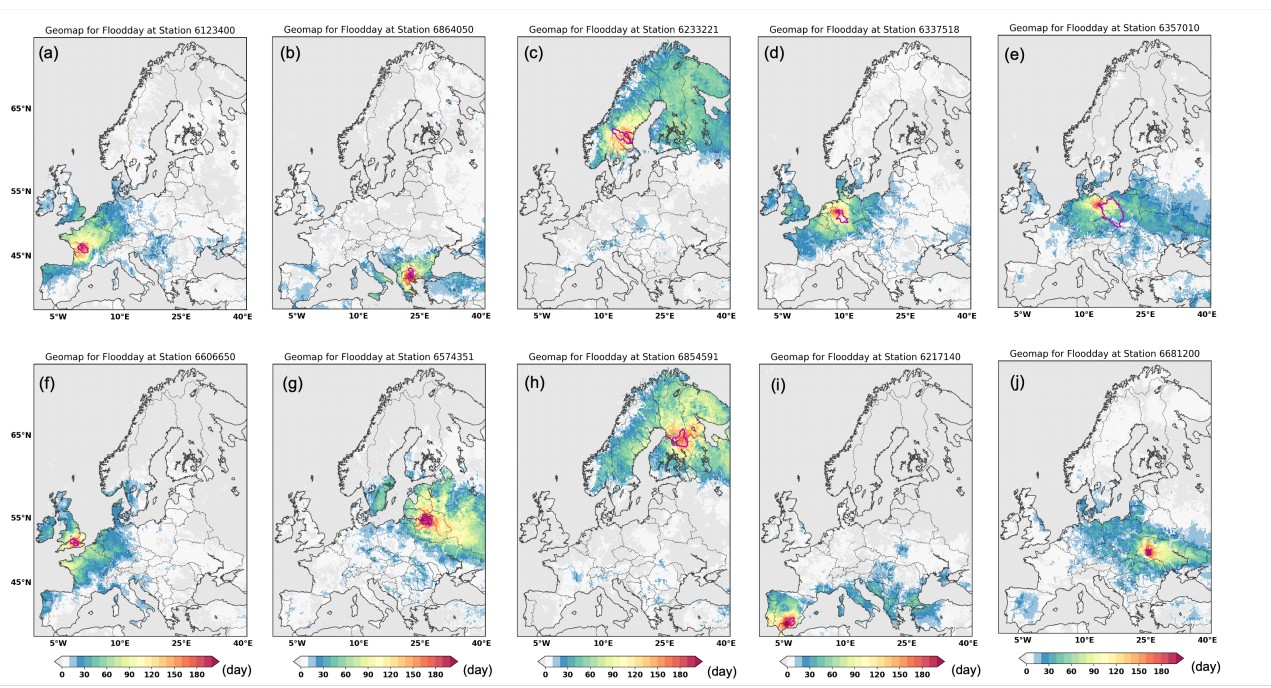

**Figure A8.** Total number of flood days conditional to days when an outlet a certain catchment (indicated in the title of the panels) is flooding. The pink polygons denote the boundary of the catchment. The total number of flood days for the outlet grids is 256 days.

*Data availability.* The mHM model routed runoff is available from the UFZ data portal (https://doi.org/10.48758/ufz.14403). The E-OBS data is available from the European Climate Assessment and Dataset (ECA&D) website at https://www.ecad.eu. The ERA5 data is available from the Copernicus Climate Data Store (https://cds.climate.copernicus.eu). The observed station runoff data collected in the paper can be requested from the Global Runoff Data Centre (GRDC), Federal Institute of Hydrology, Koblenz, Germany, at https://www.bafg.de/GRDC/EN/

415 Home/homepage_node.html. The GloFAS-ERA5 river discharge reanalysis is provided through the European Commission Copernicus Emergency Management Service (CEMS) and can be downloaded from https://cds.climate.copernicus.eu/cdsapp#!/dataset/cems-glofas-historical. Historical population extracted from GPWv4 data can be downloaded from https://sedac.ciesin.columbia.edu/data/collection/gpw-v4.

*Author contributions.* BJ and JZ conceived the study. BJ performed all analyses and wrote the initial draft. EB contributed to the interpretation of the results. OR conducted the mHM simulations and helped with the interpretation. All authors substantially contributed to the final

420 draft.

*Competing interests.* The corresponding author has declared that none of the authors have any competing interests.

*Acknowledgements.* BF and JZ acknowledge funding from the Helmholtz Initiative and Networking Fund (Young Investigator Group COM-

425 POUNDX, Grant Agreement VH-NG-1537). EB and JZ acknowledge the European Union's Horizon 2020 research and innovation program within the project 'XAIDA: Extreme Events - Artificial Intelligence for Detection and Attribution' under grant agreement No 101003469. EB received funding from the DFG via the Emmy Noether Programme (grant ID 524780515).

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
