# Peer review of "An increase in the spatial extent of European floods over the last 70 years"

_EGUsphere, 2023_

## Author Comment (AC3)

The manuscript titled "An increase in the spatial extent of European floods over the last 70 years" by Fang et al., aims to identify large spatio-temporally connected flood events over the last 70 years in Europe using the mHM hydrological model.

Overall, the manuscript is well organized and written. For some of the methods used in the study a better justification should be presented so that the reader is better aware of the reasons for certain choices.

For further details find my comments below.

**General Comments:**

Naturally, as with such a complex study, a lot of data sets and models have been used so that at some stage the reader might get lost what has actually been done. I think it would be helpful to add a schematic/graphical representation showing the inputs, models used and outputs and their ways of comparison in one to two panels to make things clearer.

**Response:**
Thanks for your suggestion and we will add a schematic figure (as shown below) in the manuscript.

[Figure]

Additionally, it is unclear why the period beginning with 1951 was used. The mHM model has a spin-up period of 1940-1959. Hence, I think the analysis should be conducted only after… This would also resolve the issue that as described in L 126 10 years of data are discarded (1981-1990). So the "attribution" could then be done using the two periods 1961-1990 and 1991-2020.

**Response:**

Thanks, it seems the spin-up of mHM was not explained in sufficient detail. The model's initial conditions were created in two decades, namely 1950-1959, and then reading the restart file of 1959 as an initial condition in 1940 run until 1949 (here, using pre1950 EOBS version, see https://surfobs.climate.copernicus.eu/dataaccess/access_eobs.php was used). This means that in total, 20 years were used to create steady-state conditions starting already in 1950. So, the decade

1950-1959 is not affected by the spin up anymore. We use the two time periods as in the manuscript to maximize the trend signal and make use of the full time period available.

Also, the section 2.5 on "attribution" of changes in flood events needs to be re-written. A lot of assumptions are being made in this section, but it remains illusive why certain choices are being made (suggest adding some explanations instead of just citing references that have done the same previously). Additionally, the authors should aim to add some "physical reasoning" to this mainly statistically based attribution exercise. Additionally, the attribution reminds of the use of conditional probabilities. Please elaborate/discuss why this approach has not been considered instead, i.e. what are the advantages of the current approach…

**Response:**
We agree that this section requires a more detailed explanation.

In general, the occurrence of a compound event is shaped by the multivariate probability density function (pdf) associated with the variables describing the event (François & Vrac, 2023; Singh et al., 2021; Sklar, 1973; Zscheischler & Seneviratne, 2017). According to copula theory, a multivariate pdf can be decomposed in the marginal pdfs (i.e. the pdf of the individual variables associated with the compound event) and the copula describing the dependencies among the individual variables. Therefore, the compound event changes are shaped by changes in the marginal pdfs and the dependencies (Bevacqua et al., 2021; Zscheischler & Seneviratne, 2017). We will add this part to section 2.5.

In our specific case, the marginal distributions describe the runoff at individual locations, while the dependency describes the dependency between the runoff at different locations. Accordingly, quantifying the contribution from these two sources is a common practice in compound event research as it provides insights into the origins of the changes. For instance, Bevacqua et al., (2021) applied this approach to differentiate the contribution from rainfall magnitude and rainfall spatial dependence to the extent of winter extreme rainfall. Here, we employ a similar approach to test the hypothesis that flood extent can potentially be influenced not only by changes in runoff magnitude but also by changes in the spatial dependence of high runoff events, as outlined in the introduction (Lines 54-56). We will further clarify the aspects above in the paper.

In the physical reasoning part, utilizing the contribution decomposition method mentioned above, we find that the expansion of flood extent across Northern Europe is primarily driven by the amplified runoff magnitude resulting from increased rainfall and snowmelt (Figure 9). Conversely, the increase of flood extent across central Europe is dominated by the enhanced spatial dependence of runoff extremes, attributed to more widespread heavy rainfall, as outlined in section 3.4. Thus, the decomposition method significantly aids in elucidating the underlying mechanisms behind changes in flood extent.

We agree that conditional probability is commonly used in attribution analysis. For instance, one can investigate the change of flood frequency or magnitude conditioned on different weather types for climate modes. However, in our case the approach would not help much in addressing our main research objective, namely to understand trends in the flood extent.

Finally, I am missing a discussion of the presence of human effects of the hydrology/floods through dams etc. particularly with regard to the conclusions drawn with regard to the attribution of changes in flood extent. With more and large dams that have been built in the recent decades in south of Europe a lot of water will be held back and is no longer available for flooding.

**Response:**

We did not include dams and reservoirs in the model setup, but we agree with the reviewer that it might indeed be a limiting factor, and we will acknowledge this in the revised manuscript. First, we are unaware of a database that would harmonise such information across the entire European domain over time. Additionally, reservoir modelling needs more crucial bathymetric data (information about the underwater topography), accounting for sedimentation processes, which is not available. These data are essential for connection among water elevation, surface area, and reservoir volume, which is necessary for constructing an accurate reservoir model at a large scale, including operational rules. We will newly acknowledge this limiting factor in the paper's discussion, as in reality, the available dam's/reservoir's storage for hydropower generation might influence the seasonal flow patterns and attenuate flood peaks and their impacts, and we propose this topic for future studies.

How do the authors reconcile the reality with the modelled results… Please elaborate and discuss.

**Response:**

The model results have been validated against various independent data sources. As illustrated in section 3.1, we not only compare the mHM simulated runoff with GRDC stations but also validate our identified flood events against an impact-based flood database (HANZE), showing a relatively high consistency. Hence, although our results focus on runoff extremes that may not necessarily lead to impactful floods, they still provide valuable insights applicable to reality.

I also in general agree with most of the comments of Reviewer 2, therefore I will not list similar comments/concerns in this review.

**Specific Comments:**

L 28: Suggest replacing "a future climate" with "future" as a future climate is not the only important factor.

**Response:**

Thanks for your suggestion, we will modify this in the revised manuscript.

L 66: Please add which version of E-OBS was used.

**Response:**

Thanks for your suggestion, the E-OBS version is eobs_v25e and we will add this information to the revised manuscript.

L 70-71: Please specify what percentage "most" refers to and what quantifies as "low" station density. Suggest also showing in the appendix area and stations that have been excluded from the analysis.

**Response:**

Thank you for your suggestion, we agree that this sentence lacks precision. Hence, we revise this statement into: A spatial mask is further applied to exclude catchments with headwater/contributing areas outside the Europe/E-OBS domain (Lehner et al., 2008).

L 82: Please specify what method was used to downscale the resolutions.

**Response:**

Apologies for the confusion. We aggregated (not downscaled) the population data by summing up the population at a 2.5 arc-minute resolution within each 0.125° grid cell. We will clarify this in the revised manuscript.

L 105: Please elaborate with a short sentence, why 0.4.

**Response:**

We have conducted sensitivity tests with the overlap ratio ranging from 0.3 to 0.5 and found that our results are not sensitive to the value of the overlap ratio.

L 112: Please elaborate why 1000 km2.

**Response:**

This choice is motivated by the fact that the mHM resolution is 0.125° × 0.125°, approximately corresponding to an area of 100 km². We aim to ensure that each large river grid cell has at least 10 upstream cells to facilitate the river routing procedure. Moreover, since our focus is on the climate effect on flood generation, we prioritize grid cells with larger catchment areas. In the future, if mHM has a finer resolution, we may also consider focusing on river grids with lower contributing catchment areas.

L 122: Please elaborate why 0.7.

**Response:**

This ratio was taken from Tarasova et al., (2023), who also applied this ratio to identify the snowmelt-driven floods. We have conducted sensitivity tests with this ratio ranging from 0.6 to 1, and found that our results are not sensitive to the value of this ratio.

L142: suggest replacing "well" with "satisfactory", as this is the terminology mainly used associated to the use of NSE values.

**Response:**

Thanks for your suggestion, we will revise this in the new manuscript.

L360: I´m not sure if one can conclude that "floods are more widespread in low-lying regions, such as parts than in high mountainous regions like the Alps." As the authors have a priori excluded smaller catchments which would naturally be found in the mountainous areas…

**Response:**

As illustrated in Line 110-112, we intended to exclude non-riverine floods, but not necessarily small-catchment floods. As depicted in Figure 4, the number of flood events in the high-mountainous region is slightly lower; nevertheless, there are still a number of events to support the conclusion.

Figures: In some Figures red and green colour coding is used. Please avoid using this as it is difficult for colour blind readers to discern the Figures. Suggest using "colour-blind safe" colours instead.

**Response:**

Thanks for your suggestion. We will modify the corresponding figures with "colour-blind safe" colors in the manuscript.

**Reference:**

Berghuijs, W. R., Allen, S. T., Harrigan, S., & Kirchner, J. W. (2019). Growing Spatial Scales of Synchronous River Flooding in Europe. *Geophysical Research Letters*, *46*(3), 1423–1428. https://doi.org/10.1029/2018GL081883

Bevacqua, E., Shepherd, T. G., Watson, P. A. G., Sparrow, S., Wallom, D., & Mitchell, D. (2021). Larger Spatial Footprint of Wintertime Total Precipitation Extremes in a Warmer Climate. *Geophysical Research Letters*, *48*(8), e2020GL091990. https://doi.org/10.1029/2020GL091990

François, B., & Vrac, M. (2023). Time of emergence of compound events: Contribution of univariate and dependence properties. *Natural Hazards and Earth System Sciences*, *23*(1), 21–44. https://doi.org/10.5194/nhess-23-21-2023

Jiang, S., Bevacqua, E., & Zscheischler, J. (2022). River flooding mechanisms and their changes in Europe revealed by explainable machine learning. *Hydrology and Earth System Sciences*, *26*(24), 6339–6359. https://doi.org/10.5194/hess-26-6339-2022

Lehner, B., Verdin, K., & Jarvis, A. (2008). New Global Hydrography Derived From Spaceborne Elevation Data. *Eos, Transactions American Geophysical Union*, *89*(10), 93–94. https://doi.org/10.1029/2008EO100001

Singh, H., Najafi, M. R., & Cannon, A. J. (2021). Characterizing non-stationary compound extreme events in a changing climate based on large-ensemble climate simulations. *Climate Dynamics*, *56*(5), 1389–1405. https://doi.org/10.1007/s00382-020-05538-2

Sklar, A. (1973). Random variables, joint distribution functions, and copulas. *Kybernetika*, 9(6), 449-460.

Tarasova, L., Lun, D., Merz, R., Blöschl, G., Basso, S., Bertola, M., Miniussi, A., Rakovec, O., Samaniego, L., Thober, S., & Kumar, R. (2023). Shifts in flood generation processes exacerbate regional flood anomalies in Europe. *Communications Earth & Environment*, *4*(1), Article 1. https://doi.org/10.1038/s43247-023-00714-8

Zscheischler, J., & Seneviratne, S. I. (2017). Dependence of drivers affects risks associated with compound events. *Science Advances*, *3*(6), e1700263. https://doi.org/10.1126/sciadv.1700263

---

## Author Response (AR1)

*We would like to thank all reviewers for their constructive comments to our manuscript. A point-by-point response to all comments raised follows below. Throughout the response, the reviewers' comments are presented in* black*, and our responses are in blue. The line and figure numbers in* red *correspond to those in the clean version of the revised manuscript.*

**Response to Reviewer 1**

This paper studies changes in spatial flood extents across Europe and its drivers using a state-of-the-art hydrological model. The paper reports that, on average, there has been an increase of 11.3% in flood extents across Europe over the past 70 years, with regionally varying trends and regionally varying drivers. The work is well-presented, comes across as robust (though with some of the extensive modeling dependencies and relatively low NSE scores for many regions that's hard to truly assess), addresses a relevant topic, and goes clearly beyond the earlier works on this topic (that are also well cited by the authors). I commend the authors for submitting such an organized manuscript and I do not think that my comments would further improve this paper. Well done!!

Thank you for the recognition of this paper!

**Response to Reviewer 2**

The manuscript "An increase in the spatial extent of European floods over the last 70 years" by Fang et al., presents a large-scale analysis of the spatial extent of floods and its spatial and temporal variations in the last 7 decades. The analysis is based on model simulations from the mHM model driven by observational data. The study finds that increased on average over most parts Europe and attributes its changes to changes in the magnitude or the spatial dependence of its drivers. The manuscript is well written, and the analyses and results are presented in a convincing way. Please find my comments below:

**Major comments:**
Comparison with HANZE: at best the detection rate of the 100 largest flood events flood events compared to the HANZE database is 50%. This low detection rate raises questions on the threshold used for the definition of flood events. The authors define flood day using the 99th percentile. This translates in more than 3 events in a year on average in each pixel. Many of the selected 'flood days' are therefore 'just high-flow days' (not even all annual maxima cause inundations). This could be a cause of this big discrepancy with the HANZE database and could suggest the use of a higher threshold for the definition of flood days (e.g., the 5- or 10-year flood).

**Response**:

Thank you for your suggestion. In addition to utilizing the flood-day threshold of 99% as presented in the manuscript, we also investigated thresholds of 99.7% and 99.9%. The ratios of identified flood events do not change much (the detection rate increased to 54% and 52% for 99.7% and 99.9%, respectively). It is worth noting that our analysis focuses solely on runoff extremes, without accounting for human infrastructure or other exposure. Consequently, runoff extremes may not always result in measurable impacts, thus may not be documented in impact-based flood databases such as HANZE. Moreover, employing a much higher threshold (e.g., the 5- or 10-year flood) for defining flood days would substantially reduce the sample size for studying trends in flood extremes and consequently increase uncertainty. We therefore kept using the 99th percentile as a compromise between sample size and accuracy of matching recorded flood impact.

Figure 9c and L241 "changes in soil moisture seem to play a minor role in the detected changes in flood characteristics. This is in contradiction with the findings of Blöschl et al. (2017, 2019) and Bertola et al. (2021), Tarasova et al. (2023) who find that antecedent soil moisture is relevant to explain negative trends in flood magnitudes (and shift in timing) and increase in flood-poor periods in the Mediterranean catchments. How do the results of this analysis compare to this literature? What are the reasons of this discrepancy? Are the changes in flood

extents caused by different drivers than trends in flood magnitudes and temporal clustering of floods? How is soil moisture estimated in this analysis?

**Response:**

We admit that this sentence lacks precision, and replacing the word "characteristics" with "extent" would be more appropriate. Additionally, we would like to clarify that when referring to the "minor" role of soil moisture, we intended to emphasize its relative importance compared to the dominant roles of snowmelt in Northern Europe and rainfall in Central and Southern Europe in influencing the change of flood extent. However, this does not imply that the effect of soil moisture is negligible.

The reason why soil moisture is not thoroughly discussed in this study is because it exhibits a more localized and secondary effect compared to rainfall and snowmelt. However, thanks to the reviewer's suggestion, upon closer examination of the soil moisture effect, we also find that it is significant in some regions. Therefore, we incorporated a discussion into the revised manuscript to further explain the role of soil moisture in Lines 255-261: "*For instance, in Germany and France, although soil moisture decreases, flood extent increases due to more extensive heavy rainfall, highlighting the dominant role of rainfall compared to soil moisture. However, its impact can be substantial at the regional scale. For example, in the northern UK, a strong increase in soil moisture contributes to the increase in flood extent; while in northern Iberia, the decrease in soil moisture is associated with a reduction in flood extent. These findings align with those of Blöschl et al., (2019) and Bertola et al., (2021), who revealed a similar role of soil moisture in influencing flood magnitude in these regions. Furthermore, in the Mediterranean Sea region, a decrease in flood extent is partly attributed to the decrease in soil moisture, generally aligning with the findings of Tarasova et al., (2023).*" Therefore, our results are not contradictory to previous studies; rather, we prioritize the investigation of dominant drivers over secondary and more localized factors such as soil moisture.

The separation of the contribution from runoff magnitude and runoff spatial dependence not only aims to disentangle the contribution from these two sources but also serves as a bridge to enhance our understanding of the drivers (e.g., rainfall, snowmelt, soil moisture, as illustrated in Section 3.4) of flood extent. In addition, soil moisture is an output from the mHM model, as stated in Line 80.

L346-348: only drivers occurring in the spatial area of flood events are considered. This has big implications in the attribution analysis especially for very large rivers, as contributing drivers occurring within the actual catchment area are not considered (often drivers occurring in one part of the catchment, e.g. snowmelt or rainfall, cause flooding downstream where these drivers do not necessarily occur). What are the implications on the attribution results?

**Response:**

We acknowledge the reviewer's concern regarding the potential implications of flood contributing area selection on the attribution results, and we also commented on this aspect in the Discussion (Section 3.5). For instance, in Figure A8, we illustrate the total number of flood days contingent upon days when an outlet of a specific catchment is flooding. Our analysis reveals that "*if an outlet grid cell experiences flooding, there is a high likelihood that grid cells within the corresponding catchment also experience flooding*" (Lines 372-373). This suggests that although we do not encompass the entire contributing catchment for each grid cell, our event-based findings remain credible.

We do note that this approach performs more effectively for smaller catchments compared to much larger ones. Nonetheless, considering the drainage area for every grid within each event would be highly challenging. Despite this limitation, our attribution results, such as the spatial distribution of snow-driven and rainfall-driven floods, exhibit strong consistency with previous studies utilizing different attribution methods (Jiang et al., 2022).

**Specific comments:**

L39-46: it is true that "other studies rely on observations and may miss important information due to uneven spatial distribution of stations". On the other hand, models (like the one used here – L65-72) are calibrated and validated using observations so have this intrinsic limitation too. Furthermore, models have other limitations, e.g. modelling uncertainty and resolution of simulations (in this case gridded runoff simulations that are quite coarse – 0.125° =~ 11km …). Furthermore, for the attribution analysis the study relies on EOBS (L79-80).

**Response**:

We agree that also hydrological models may be affected by the uneven distribution of runoff stations. However, it is important to highlight that validating grid-based simulations against station-based observations implies not only the model's feasibility and reliability for the validated locations but also the potential feasibility of other unvalidated locations due to the physical processes encoded into the model. This aspect can provide valuable insights into ungauged areas and improve the quantification of flood extent.

Additionally, despite the model's resolution limitation (e.g., 0.125° in our study), it still allows for a more refined estimation of flood extent compared to station-based research. For example, the flood synchrony scale, commonly used for quantifying flood extent from a station-based perspective, is defined as the largest radius of the circle within which half of the station flooding occurs near simultaneously (Berghuijs et al., 2019). However, its application in regions with scarce runoff stations may introduce considerable bias due to the distance between stations. While the grid-based simulation could help alleviate this limitation. Note that we also acknowledge other limitations of the model simulation besides resolution, as illustrated in Lines 349-356.

L21 "spatially compounding river floods": the use of "compounding" seems inappropriate in this context as we are talking about spatially widespread events and there is no mention of simultaneous occurrence of flood drivers in this section (or elsewhere in the paper). I suggest substituting "compounding" with "widespread" or similar.

**Response**:

Thanks for your suggestion. We changed it in the revised manuscript (Line 21).

L52: I suggest citing Lun et al. (2020) – the first study detecting flood-rich and flood-poor periods in Europe.

**Response**:

Thanks for your suggestion. We added it in the manuscript (Line 52).

L100-106: are flood days defined at each grid cell separately? Spatially connected flood days (i.e. pixels) and overlapping flood patches are 'further combined'. Does this mean that they are considered as one single event? Please clarify.

**Response**:

Yes, the flood days are defined at each grid cell separately and then the overlapping flood patches are further combined together to form a single event. We clarified this part in the revised manuscript (Line 108).

L125: E_past and E_pres denote the AVERAGE flood extent?

**Response**:

Yes, E_past and E_pres denote the temporally-averaged flood extent over the historical period (1951-1980) and present period (1991–2020) for each grid cell, respectively. We clarified this in the revised manuscript (Line 137).

L158: 244 flash floods from the HANZE dataset are captured in the dataset. However, flash floods typically occur over small catchments (a few km2) while the catchment area that is captured is at least of the order of magnitude of 1000 km2 (due to the resolution of the gridded simulations). Similarly for the time dimension, i.e. flash floods typically last less than 24h, while the resolution of the simulations is daily. How can mHM model simulations capture such flash floods? What are the implications in terms of such identified events?

**Response**:

Yes, we acknowledge that flash floods often have relatively small spatial extents. However, note that in our comparison, we assess the identified flood extent against the affected NUTS3 regions documented in the HANZE database, rather than pinpointing the exact spatial extent of specific flash floods. This approach may potentially increase the detection ratio of flash floods.

Regarding temporal resolution, as outlined in the method section, our detection method employs a moving window of ±3 days (Line 97). Additionally, the temporal resolution of the impact-based flood dataset (HANZE) is also daily, which helps in capturing short-duration flash floods. The implication of identifying such events is to provide confidence in the applicability of mHM simulations and the flood detection algorithm for studying changes in large-scale floods.

Figure 2: is this figure only representing the events in the HANZE dataset or does it contain results of this analysis? If it does not contain results of this analysis, it should be moved to another section (e.g. appendix).
**Response**:
Yes, thanks for your suggestion. The figure only represents the events in the HANZE dataset, and we have put it in the appendix section (Figure A2).

Figure 9: it is not clear if the maps show changes in snowmelt, rainfall and soil moisture over all days in the two periods or if they refer to changes for flood events only (i.e. only the rainfall causing flood events or changes in rainfall in general?)
**Response**:
Sorry for the confusion. The spatial maps in Figure 9 show changes in snowmelt, rainfall and soil moisture over all days in the two periods. We have clarified this in the revised manuscript.

L329: "aligns closely with an independent impact-based
**Response**:
We're not sure what this comment refers to.

Figure A1: labels and titles of the plots are not fully clear and not explained in the caption.
**Response**:
Thanks, we provided more details for Figure A1 in the revised manuscript as follows: *"Specifically, "MAF" compares observed and simulated maximum annual floods in terms of their discharge and day of occurrence; "MAF date" compares the observed and simulated discharge on the exact date of the observed annual floods; "MAF event" compares observed discharge and date of maximum annual flood to the peak of corresponding runoff event. More details about these three cases can be found in Tarasova et al., 2023."*

**Response to Reviewer 3**

The manuscript titled "An increase in the spatial extent of European floods over the last 70 years" by Fang et al., aims to identify large spatio-temporally connected flood events over the last 70 years in Europe using the mHM hydrological model.

Overall, the manuscript is well organized and written. For some of the methods used in the study a better justification should be presented so that the reader is better aware of the reasons for certain choices.

**Response:**
Thanks for the feedback, we now provide a schematic overview figure and justifications for the different methodological choices.

For further details find my comments below.

**General Comments:**

Naturally, as with such a complex study, a lot of data sets and models have been used so that at some stage the reader might get lost what has actually been done. I think it would be helpful to add a schematic/graphical representation showing the inputs, models used and outputs and their ways of comparison in one to two panels to make things clearer.

**Response:**
Thanks for your suggestion, we added a schematic figure (Figure 1) in the revised manuscript as follows:

[Figure]

**Figure 1**. Main workflow of the study. The figure outlines the main analysis steps undertaken in this paper. In addition to the mHM model simulations driven by E-OBS data, we also analyze and compare the results obtained against mHM simulation driven by ERA5 data and the GloFas dataset.

Additionally, it is unclear why the period beginning with 1951 was used. The mHM model has a spin-up period of 1940-1959. Hence, I think the analysis should be conducted only after… This would also resolve the issue that as described in L 126 10 years of data are discarded (1981-1990). So the "attribution" could then be done using the two periods 1961-1990 and 1991-2020.

**Response:**

Thanks, it seems the spin-up of mHM was not explained in sufficient detail. The model's initial conditions were created in two decades, namely 1950-1959, and then reading the restart file of 1959 as an initial condition in 1940 run until 1949 (here, using pre1950 EOBS version, see https://surfobs.climate.copernicus.eu/dataaccess/access_eobs.php was used). This means that

in total, 20 years were used to create steady-state conditions starting already in 1950. So, the decade 1950-1959 is not affected by the spin up anymore. We use the two time periods as in the manuscript to maximize the trend signal and make use of the full time period available. We added the spin-up details in the revised manuscript as follows: "*For the spin-up of mHM, the model was firstly initialized in 1950-1959, and then the restart file of 1959 was read as an initial condition in 1940 run until 1949. This means that in total, 20 years were used to create steady-state conditions starting in 1950*" (Lines 70-72).

Also, the section 2.5 on "attribution" of changes in flood events needs to be re-written. A lot of assumptions are being made in this section, but it remains illusive why certain choices are being made (suggest adding some explanations instead of just citing references that have done the same previously). Additionally, the authors should aim to add some "physical reasoning" to this mainly statistically based attribution exercise. Additionally, the attribution reminds of the use of conditional probabilities. Please elaborate/discuss why this approach has not been considered instead, i.e. what are the advantages of the current approach…

**Response:**

We agree that this section requires a more detailed explanation. We added the explanation in Lines 128- 136: "*In general, the occurrence of a compound event is shaped by the multivariate probability density function (pdf) associated with the variables describing the event (François & Vrac, 2023; Singh et al., 2021; Sklar, 1973.; Zscheischler & Seneviratne, 2017). According to copula theory, a multivariate pdf can be decomposed into the marginal pdfs and the copula describing the dependence between the individual variables (Sklar, 1973). Therefore, compound event changes can be shaped by changes in the marginal pdfs and changes in the dependence structure (Bevacqua et al., 2021; Zscheischler & Seneviratne, 2017). In our specific case, the marginal distributions describe the runoff at individual locations, while the dependence describes the dependence between the runoff at different locations. Quantifying the contribution to the compound event changes from marginal distribution and dependencies is common in compound event research as it provides insights into the origins of changes (Bevacqua et al., 2021; François & Vrac, 2023; Zscheischler & Seneviratne, 2017).*" For instance, Bevacqua et al., (2021) applied this approach to differentiate the contribution from rainfall magnitude and rainfall spatial dependence to the extent of winter extreme rainfall. Here, we employ a similar approach to test the hypothesis that flood extent can potentially be influenced not only by changes in runoff magnitude but also by changes in the spatial dependence of high runoff events, as outlined in the introduction (Lines 53-56).

In the physical reasoning part, utilizing the contribution decomposition method mentioned above, we find that the expansion of flood extent across Northern Europe is primarily driven by the amplified runoff magnitude resulting from increased rainfall and snowmelt (Figure 9).

Conversely, the increase of flood extent across central Europe is dominated by the enhanced spatial dependence of runoff extremes, attributed to more widespread heavy rainfall, as outlined in Section 3.4. Thus, the decomposition method strongly aids in elucidating the underlying mechanisms behind changes in flood extent.

We agree that conditional probability is commonly used in attribution analysis. For instance, one can investigate the change of flood frequency or magnitude conditioned on different weather types for climate modes. However, in our case the approach would not help much in addressing our main research objective, namely to understand trends in the flood extent and its drivers.

Finally, I am missing a discussion of the presence of human effects of the hydrology/floods through dams etc. particularly with regard to the conclusions drawn with regard to the attribution of changes in flood extent. With more and large dams that have been built in the recent decades in south of Europe a lot of water will be held back and is no longer available for flooding.

**Response:**
We did not include dams and reservoirs in the model setup, but we agree with the reviewer that it might indeed be a limiting factor, and we acknowledged this in the revised manuscript (Lines 353-356):*"Furthermore, we note that no information on dam and reservoir constructions, which might change flood dynamics over time, are included in the model setup. We are unaware of a database that would harmonise such information across the entire European domain over time. Possibly even more crucial, bathymetric data, which would be needed for sedimentation processes, is not available."*

How do the authors reconcile the reality with the modelled results… Please elaborate and discuss.

**Response:**
The model results have been validated against various independent data sources. As illustrated in Section 3.1, we not only compare the mHM simulated runoff with GRDC stations but also validate our identified flood events against an impact-based flood database (HANZE), showing a relatively high consistency. Hence, although our results focus on runoff extremes that may not necessarily lead to impactful floods, they still provide valuable insights applicable to reality.

I also in general agree with most of the comments of Reviewer 2, therefore I will not list similar comments/concerns in this review.

**Specific Comments:**

L 28: Suggest replacing "a future climate" with "future" as a future climate is not the only important factor.

**Response:**
Thanks for your suggestion, we modified this in the revised manuscript (Line 28).

L 66: Please add which version of E-OBS was used.

**Response:**
Thanks for your suggestion, the E-OBS version is version 25.0e and we added this information to the revised manuscript (Line 66).

L 70-71: Please specify what percentage "most" refers to and what quantifies as "low" station density. Suggest also showing in the appendix area and stations that have been excluded from the analysis.

**Response:**
Thank you for your suggestion, we agree that this sentence lacks precision. Hence, we revised this statement into: "*A spatial mask is further applied to exclude catchments with headwater/contributing areas outside the Europe/E-OBS domain (Lehner et al., 2008)*" (Lines 72 - 73).

L 82: Please specify what method was used to downscale the resolutions.

**Response:**
Apologies for the confusion. We aggregated (not downscaled) the population data by summing up the population at a 2.5 arc-minute resolution within each 0.125° grid cell. We clarified this in the revised manuscript (Lines 83-84).

L 105: Please elaborate with a short sentence, why 0.4.

**Response:**
We have conducted sensitivity tests with the overlap ratio ranging from 0.3 to 0.5 and found that our results are not sensitive to the value of the overlap ratio. We have added this information to the revised manuscript (Line 108).

L 112: Please elaborate why 1000 km2.

**Response:**
This choice is motivated by the fact that the mHM resolution is $0.125° \times 0.125°$, approximately corresponding to an area of 100 km$^2$. We aim to ensure that each large river grid cell has at

least 10 upstream cells to facilitate the river routing procedure. Moreover, since our focus is on the climate effect on flood generation, we prioritize grid cells with larger catchment areas. In the future, if mHM has a finer resolution, we may also consider focusing on river grids with lower contributing catchment areas.

L 122: Please elaborate why 0.7.

**Response:**
This ratio was taken from Tarasova et al., (2023), who also applied this ratio to identify the snowmelt-driven floods. We have conducted sensitivity tests with this ratio ranging from 0.6 to 1, and found that our results are not sensitive to the value of this ratio. We have added this information to the revised manuscript (Lines 124-125).

L142: suggest replacing "well" with "satisfactory", as this is the terminology mainly used associated to the use of NSE values.

**Response:**
Thanks for your suggestion, we revised this in the new manuscript (Line 155).

L360: I'm not sure if one can conclude that "floods are more widespread in low-lying regions, such as parts than in high mountainous regions like the Alps." As the authors have a priori excluded smaller catchments which would naturally be found in the mountainous areas…

**Response:**
As illustrated in Lines 112-114, we intended to exclude non-riverine floods, but not necessarily small-catchment floods. As depicted in Figure 4, the number of flood events in the high-mountainous region is slightly lower; nevertheless, there are still a number of events to support the conclusion.

Figures: In some Figures red and green colour coding is used. Please avoid using this as it is difficult for colour blind readers to discern the Figures. Suggest using "colour-blind safe" colours instead.

**Response:**
Thanks for your suggestion. We modified the corresponding figures with "colour-blind safe" colors in the manuscript.

**Reference**

Berghuijs, W. R., Allen, S. T., Harrigan, S., & Kirchner, J. W. (2019). Growing Spatial Scales of Synchronous River Flooding in Europe. *Geophysical Research Letters*, *46*(3), 1423–1428. https://doi.org/10.1029/2018GL081883

Bertola, M., Viglione, A., Vorogushyn, S., Lun, D., Merz, B., & Blöschl, G. (2021). Do small and large floods have the same drivers of change? A regional attribution analysis in Europe. *Hydrology and Earth System Sciences*, *25*(3), 1347–1364. https://doi.org/10.5194/hess-25-1347-2021

Bevacqua, E., Shepherd, T. G., Watson, P. A. G., Sparrow, S., Wallom, D., & Mitchell, D. (2021). Larger Spatial Footprint of Wintertime Total Precipitation Extremes in a Warmer Climate. *Geophysical Research Letters*, *48*(8), e2020GL091990. https://doi.org/10.1029/2020GL091990

Blöschl, G., Hall, J., Viglione, A., Perdigão, R. A. P., Parajka, J., Merz, B., Lun, D., Arheimer, B., Aronica, G. T., Bilibashi, A., Boháč, M., Bonacci, O., Borga, M., Čanjevac, I., Castellarin, A., Chirico, G. B., Claps, P., Frolova, N., Ganora, D., … Živković, N. (2019). Changing climate both increases and decreases European river floods. *Nature*, *573*(7772), Article 7772. https://doi.org/10.1038/s41586-019-1495-6

François, B., & Vrac, M. (2023). Time of emergence of compound events: Contribution of univariate and dependence properties. *Natural Hazards and Earth System Sciences*, *23*(1), 21–44. https://doi.org/10.5194/nhess-23-21-2023

Jiang, S., Bevacqua, E., & Zscheischler, J. (2022). River flooding mechanisms and their changes in Europe revealed by explainable machine learning. *Hydrology and Earth System Sciences*, *26*(24), 6339–6359. https://doi.org/10.5194/hess-26-6339-2022

Lehner, B., Verdin, K., & Jarvis, A. (2008). New Global Hydrography Derived From Spaceborne Elevation Data. *Eos, Transactions American Geophysical Union*, *89*(10), 93–94. https://doi.org/10.1029/2008EO100001

Singh, H., Najafi, M. R., & Cannon, A. J. (2021). Characterizing non-stationary compound extreme events in a changing climate based on large-ensemble climate simulations. *Climate Dynamics*, *56*(5), 1389–1405. https://doi.org/10.1007/s00382-020-05538-2

Sklar, A. (n.d.). *Random Variables, Joint Distribution Functions, and Copulas*.

Tarasova, L., Lun, D., Merz, R., Blöschl, G., Basso, S., Bertola, M., Miniussi, A., Rakovec, O., Samaniego, L., Thober, S., & Kumar, R. (2023). Shifts in flood generation processes exacerbate regional flood anomalies in Europe. *Communications Earth & Environment*, *4*(1), Article 1. https://doi.org/10.1038/s43247-023-00714-8

Zscheischler, J., & Seneviratne, S. I. (2017). Dependence of drivers affects risks associated with compound events. *Science Advances*, *3*(6), e1700263. https://doi.org/10.1126/sciadv.1700263

---

## Author Response (AR2)

*We would like to thank all reviewers for their constructive comments on our manuscript. A point-by-point response to all comments raised follows below. Throughout the response, the reviewers' comments are presented in* black*, and our responses are in blue. The line and figure numbers in* red *correspond to those in the clean version of the revised manuscript.*

**Response to Reviewer 2**

I thank the authors for their reply. However, I am not fully satisfied by their answer. Please find my comments below:

1) Comparison with HANZE: based on the answer of the authors I argue that the choice of comparing the runoff extremes with HANZE events is reasonable/meaningful. This is because in the HANZE database only floods that had big impacts are reported and this reasonably correspond to a flood that is at least a flood associated with a 10-year return period which is much larger than your detection threshold (even the threshold 99.9% of daily runoff is not really a 'big flood'). From my point of view you are comparing two very different things and you should use a different observational database to 'validate' you event selection. Also, I do not agree with what stated at L358-360 ("Furthermore, the resulting flood event database aligns closely with an independent impact-based flood record database […]"). If this sentence refers to the HANZE (which I am not 100% sure) I would not define a 50% detection rate as "close alignment".

Response: We acknowledge the reviewer's concern regarding the differences between the two flood datasets. For a physically based evaluation of runoff, we provide Figure 2. In Figure 3, we assess whether the identified top events via the algorithm are impactful. We agree that flood events identified in this study using extreme runoff (i.e., 99th percentile) do not necessarily correspond to impactful floods as those documented in the HANZE database, not least because we might detect floods in areas with little exposure. We thus concur that the HANZE dataset may not be suitable for formal validation of our dataset due to significant differences in their severity levels.

However, we argue that impactful flood events recorded in the HANZE database should, in theory, be detectable by a robust flood detection algorithm, also when a relatively moderate detection threshold is used. Our results indicate that approximately 75% of river floods documented in HANZE are detected by our algorithm, thereby affirming that our approach captures impactful floods (besides many other floods). Moreover, we selected the HANZE dataset for comparison because it includes detailed information on the regions affected by floods, which facilitates a more direct comparison with the flood extent areas identified by our 3D detection algorithm.

In summary, rather than using the HANZE dataset for a formal validation of our database we use it to assess whether we are able to capture societally-relevant, impactful events through our flood detection algorithm. To avoid confusion, we now added to the manuscript "*Note that the HANZE dataset is not used for formal validation of our flood events due to the large differences in the severity levels of floods between these two datasets. Instead, it serves more as a comparison to assess our flood detection algorithm's adequacy, particularly in capturing impactful floods, even with a relatively moderate detection threshold*" in Lines 99-102.

Regarding L358-360 this statement primarily refers to our findings that our algorithm detects about 75% of the river floods recorded in HANZE. We acknowledge the need for clarity in this aspect and revised this point in the new manuscript by saying "*the resulting flood event database aligns closely with an independent impact-based flood record database by capturing 75% of recorded impactful river floods, ...*" in Lines 351-352.

2) Soil moisture effect: I do understand the general reasoning. However, why do the authors state in their reply that the effect of soil moisture is 'more localized'? I could not deduce it from the results/figures presented. By reasoning I would rather assume the opposite (i.e. soil moisture is usually changing gradually and more smoothly in space than e.g. extreme precipitation that can be very localized). Please justify your statement.

Response: We apologize for the confusion. In our previous response, the term "more localized" was used to emphasize an aspect of our findings, that is that the significance of soil moisture varies by region, while precipitation consistently plays a more dominant role in inducing floods across central and southern Europe.

3) only drivers occurring in the spatial area of flood events are considered. I accept the answers of the authors, but I think this is an important limitation that is not really clearly stated in the manuscript. L380-381 reads a bit vague and not accurate. E.g. what is "nearby precipitation"? It would be more correct to say that the precipitation and snow "falling within the corresponding hydrological catchment is not considered". Also, "spatial area of the flood events" is a bit vague. "Same pixel" would read more precise.

Response Thank you for your suggestion. We have clarified our statement in the revised manuscript by specifying that "*the precipitation and snowmelt falling outside the event but within the corresponding hydrological catchment are not considered*" in Lines 374-375. Additionally, we have modified the sentence to "*within the spatial area, that is, the same pixels of the flood events*" for greater precision in Lines 373-374.

**Response to Reviewer 3**

Overall, my previously raised points and concerns have been appropriately addressed in the revised version of the manuscript and it has improved a lot in clarity.

Some points that I had not noticed before should still be improved to improve clarify and further strengthen the document before I can recommend publication.

L 86: Please specify how snowmelt and soil moisture were derived from mHM. I.e. what quantities are used, what qualifies as such process. E.g. temperature thresholds used, how is it being checked if there is actual snow accumulation before the melting event, etc.

As this will be important for the interpretation of the rest of the study.

This will probably also help clarify the following point

Response: The snowmelt derived from mHM is based on the day-degree method, with a model-specific temperature threshold parameter (*snowTreshholdTemperature* = 1.32°C) used in the presented European setup. At an hourly model's internal time step, depending on this parameter, either snow accumulation or snowmelt occurs. Please refer to the model's code for the snow accumulations under this link:

https://git.ufz.de/mhm/mhm/-/blob/develop/src/mHM/mo_snow_accum_melt.f90?ref_type=heads#L133

As you can see from the model's equations, snow accumulation can happen before snowmelt, but we don't distinguish between these two types. As you can further see, the rate of snowmelt is driven by another parameter *degreeDayFactor*. We also further note that *degreeDayFactor* is separately defined for the three major land use classes (i.e., forest, impervious and pervious). The model's equation of mHM are described in earlier studies (Samaniego, et al. 2010; doi: 10.1029/2008WR007327 and Telteu et al. 2021; doi: 10.5194/gmd-14-3843-2021) and model's open source code is available at git repository (see https://git.ufz.de/mhm/mhm), where details on the soil moisture can be obtained. In short, soil moisture is represented by 6 layers/buckets, corresponding to the SoilGrids profiles (i.e. 5cm, 15cm, 30cm. 50cm, 100cm, 200cm), which is being filled by effective precipitation (rainfall and/or snowmelt), from where upward evapotranspiration flux happens depending on the degree of saturation of the soil and evaporative demand and consequently downward infiltration flux to unsaturated zone is estimated, which

further drains the soil storage. More details are provided in Samaniego, et al. 2010; doi: 10.1029/2008WR007327 and Telteu et al. 2021; doi: 10.5194/gmd-14-3843-2021.

In our previous manuscript, we estimated snowmelt by calculating the difference in snowpack between two consecutive days, as the earlier version of the mHM model did not provide snowmelt data directly. However, in the latest version of mHM, snowmelt is now a direct output. We have updated all relevant results using this new snowmelt output, as shown in Figure 5, Figure 6 and Figure 9a. As anticipated, the updated results are highly consistent with the previous ones, demonstrating improved accuracy.

L 271: I find Figure 9 a confusing. However, based on the outcome of this map I´m not sure if the method used is appropriate… Please carefully check the results…

a) The map is showing significant reductions of snowmelt in areas where there is basically never snow… e.g. in southern Europe

Response: We appreciate the reviewer's observation regarding Figure 9a. In the current depiction, the map includes all grid cells, regardless of their frequency of snowfall. If we adjust the criteria to only include areas with over 30 days of snowmelt larger than 2 mm in both the earlier (1951-1980) and present (1991-2000) periods, the resulting map indeed omits some regions of southern Europe, predominantly retaining the mountainous areas (as shown in the figure below). We therefore have updated the manuscript using this figure.

[Figure]

b) around the Alqueva dam, the largest artificial lake in Europe, the map is showing an increase…

In that region the winter temperatures are so high that there is practically no snow accumulation and with the lake the temperatures are even higher…

This is further raising the question of how/on what basis an increase/decrease in snowmelt being calculated.

Response: Similar to the previous response, the observed increase in snowmelt at the Alqueva dam is statistically insignificant. As illustrated in the figure above, if we apply a criterion of over 30 days of snowmelt larger than 2 mm, this region does not meet the threshold and is therefore excluded from our analysis. We have included this revised figure in the manuscript to help clarify and resolve the confusion.

Maybe, to help with the interpretation, apart of a detailed explanation of how snowmelt and soil moisture were obtained/derived, it would be good to have on Fig 4 some sort of delineation that would indicate in which areas there is more than a zero fraction of snowmelt/rainfall driven floods as shown in Figure 5.

Response: We think that such a delineation would create more confusion than clarity as it would overload the figures and double some of the information in Figure 5. We hope the explanations above and the updated figures are clear enough on this matter.